# Complex chromosomal neighborhood effects determine the adaptive potential of a gene under selection

**Magdalena Steinrueck, Călin C Guet\***

Institute of Science and Technology Austria, Klosterneuburg, Austria

**Abstract** How the organization of genes on a chromosome shapes adaptation is essential for understanding evolutionary paths. Here, we investigate how adaptation to rapidly increasing levels of antibiotic depends on the chromosomal neighborhood of a drug-resistance gene inserted at different positions of the *Escherichia coli* chromosome. Using a dual-fluorescence reporter that allows us to distinguish gene amplifications from other up-mutations, we track in real-time adaptive changes in expression of the drug-resistance gene. We find that the relative contribution of several mutation types differs systematically between loci due to properties of neighboring genes: essentiality, expression, orientation, termination, and presence of duplicates. These properties determine rate and fitness effects of gene amplification, deletions, and mutations compromising transcriptional termination. Thus, the adaptive potential of a gene under selection is a system-property with a complex genetic basis that is specific for each chromosomal locus, and it can be inferred from detailed functional and genomic data.

## Introduction

In the process of regulatory evolution, a finite set of genes are continuously combined to form new gene expression patterns and create a myriad of phenotypes (*Carroll, 2000*; *Wittkopp et al., 2004*; *Wray, 2007*). Acquiring mutations that increase the expression of a single gene can be sufficient to make an individual substantially fitter than its competitors. For example, increased expression of drug target or efflux genes is a common mechanism for the evolution of resistance to antibiotics (*Li et al., 2015*; *Palmer and Kishony, 2014*), chemotherapeutics (*Cole et al., 1992*), and insecticides (*Devonshire and Field, 1991*; *Coderre et al., 1983*). Increased expression of individual genes also provides access to new nutrient resources (*Notebaart et al., 2014*) and tolerance to diverse toxins (*Soo et al., 2011*). The fitness effect of increased expression of individual genes has mostly been determined in plasmid-based overexpression libraries (*Notebaart et al., 2014*; *Soo et al., 2011*). However, the large majority of genes reside on chromosomes, neighboring other genes, and thus mutations affecting gene expression occur in a specific chromosomal context. Unequal mutation rates along the genome (*Foster et al., 2013*; *Anderson and Roth, 1981*) imply that the chromosomal location can affect the adaptive potential of a gene, that is, the probability that adaptive mutations increasing expression of the gene will spread in a population under given selective conditions.

Adaptation by increased gene expression can result from mutations of different types (*Blank et al., 2014*; *Lind et al., 2015*): point mutations, promoter insertion by mobile elements (*Mahillon and Chandler, 1998*; *Ellison and Bachtrog, 2013*; *Stoebel et al., 2009*), promoter capture by chromosomal rearrangements (*ar-Rushdi et al., 1983*; *Blount et al., 2012*; *Xiao et al., 2008*), and gene duplication or amplification, which increases expression by way of gene dosage (*Andersson and Hughes, 2009*; *Elliott et al., 2013*). How the rate of mutation of these individual

**\*For correspondence:** calin@ist.ac.at

**Competing interests:** The authors declare that no competing interests exist.

mutation types depends on chromosomal position has in part been determined experimentally (*Foster et al., 2013*; *Hudson et al., 2002*; *Mahillon and Chandler, 1998*; *Craig, 1997*; *Touchon et al., 2009*; *Anderson and Roth, 1981*; *Seaton et al., 2012*; *Wahl et al., 1984*). Despite considerable experimental data, we currently lack an understanding of how position biases of the different mutation types together combine across different chromosomal loci, and therefore, how the chromosomal context of a gene under selection affects overall adaptation.

Here, we investigate how the complex interplay of different mutation types and mutation rate biases gives rise to an effect of chromosome position on adaptation in *Escherichia coli*. To this end, we use a single chromosomal drug resistance gene as the target of selection and a two-color fluorescence reporter readout for adaptive mutations in evolution experiments. We quantify the effect of the chromosomal position of the selected gene on adaptation and identify the mutation types underlying this effect. We find that a strong effect of chromosome position on adaptation is largely explained by rate differences of gene duplications and fitness effect differences of two types of promoter co-opting mutations (promoter capture deletions and mutations that cause read-through across upstream transcriptional terminators). Both the observed rate differences and fitness effect differences depend on simple features of the chromosomal neighborhood of the gene under selection. This suggests that the adaptive potential of a gene can be estimated by looking for respective features of chromosomal neighborhoods in genomics data. Based on these results, we propose that the chromosomal context of a gene under selection is an important factor in adaptation.

## Results

### A dual-fluorescence reporter cassette for tracking the dynamics of adaptive mutations of different types

We devised an evolution experiment with *E. coli,* in which we use a single target of selection embedded in a genetic cassette that serves as a reporter of adaptive potential and mutation types. The reporter cassette can be inserted at any chromosomal position (*Figure 1A* and *Figure 1B*), and it allows us to distinguish amplifications from other adaptive mutations in real-time using two-color fluorescence measurements. The reporter cassette contains a promoterless, translational *tetA-yfp* gene fusion followed by a transcriptional terminator and a constitutively expressed *cfp* gene. Mutations that increase expression of the tetracycline efflux pump TetA-YFP can be selected with antibiotic and monitored through YFP fluorescence (*Figure 1C*, left). Due to the immediate proximity of the *tetA-yfp* and *cfp* genes, the large majority of *tetA-yfp* amplifications are expected to contain the *cfp* gene as well. Thus, adaptation by reporter cassette amplification is expected to be distinguishable from other up-mutations by a fluorescence increase of both YFP and CFP (*Figure 1C*, right). We integrated the reporter cassette at four different intergenic loci (*A, B, C,* and *D*) along the chromosome of an *E. coli ΔtolC* strain (*Figure 1A*), giving rise to four strains (strain *A, B, C,* and *D*). The four loci were chosen to lie in intergenic regions between divergently transcribed genes in order to exclude transcription from upstream genes into the *tetA-yfp* gene (*Figure 1B*). Loci *A* and *C* are located approximately in the middle of the right and left replichore, respectively. Since we wanted to also include a locus close to the origin of replication, where no pair of divergently oriented genes is present, we chose a locus in the relatively large intergenic region between the co-oriented *rsmG* and *atpI* genes (locus *D*, *Figure 1B*), a locus previously used for large insertions (*Kuhlman and Cox, 2010*). Locus *B* was chosen based on its vicinity to several insertion sequences (IS).

We used a *ΔtolC* genetic background in order to constrain the spectrum of possible adaptive mutations to the reporter cassette locus. TolC is an outer membrane porin and an essential part of several *E. coli* multi-drug efflux pumps, which are a frequent target of selection during drug exposure (*Li et al., 2015*) and which cause low-level intrinsic resistance of *E. coli* to tetracyclines (*Sulavik et al., 2001*). By employing daily increasing levels of tetracycline (*Figure 1D*) and constant daily dilution, we created an experimental evolutionary rescue scenario (*Carlson et al., 2014*), in which populations of ancestral cells rapidly undergo extinction. Rescue from extinction requires the spread of adaptive mutations activating *tetA-yfp* expression in a race against population decline.

The probability of evolutionary rescue depends on the size and decline rate of an unadapted population, and on a combination of rate and fitness effect of adaptive mutations (*Martin et al., 2013*). We chose selective conditions such that the initial population size and decline rate are approximately

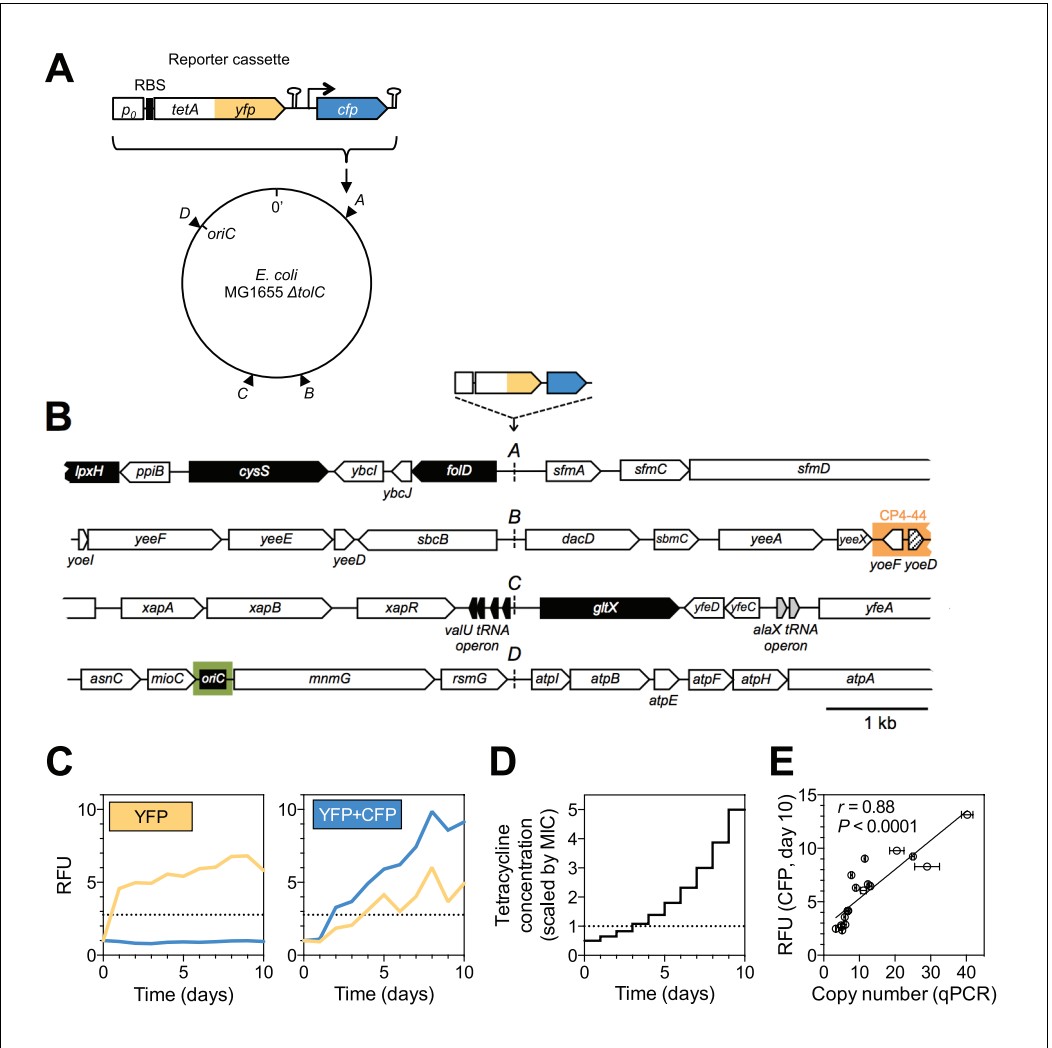

**Figure 1.** A dual-fluorescence reporter cassette for real-time tracking of adaptive mutations of different types. (**A**) Reporter cassette construct for chromosomal insertion. $p_0$ = 188 bp random DNA sequence, RBS = ribosomal binding site, hairpins = transcriptional terminators, *tetA-yfp* = selected gene, *cfp* = constitutively expressed amplification reporter. *A, B, C, D* = intergenic chromosomal insertion loci, *oriC* = origin of replication. (**B**) Immediate chromosomal neighborhoods of loci *A-D*. Black arrows = essential genes. White arrows = non-essential genes. Grey arrows = no essentiality data available. Patterned arrow (*yoeD*) = pseudogene. Orange = cryptic prophage CP4-44. Green = origin of replication (*oriC*). Chromosomal neighborhoods of loci *B*, *C*, and *D* are shown reversed with respect to conventional chromosome coordinates, so that the orientation relative to the reporter cassette is shown in the same way for all four loci. Reporter cassette genes are not drawn to scale. (**C**) Example fluorescence trajectories of rescued populations with YFP or YFP+CFP (amplification) fluorescence phenotype. RFU = relative fluorescence units (see Methods), yellow and blue lines = YFP and CFP fluorescence, dotted lines = threshold for phenotype classification. (**D**) Increase of tetracycline concentration in ten-day experiment, normalized to strain-specific minimal inhibitory concentration (MIC, dotted line). (**E**) qPCR validation of CFP fluorescence as an indicator of extent of amplifications. x-axis: *tetA-yfp* copy number as determined by qPCR on genomic DNA of rescued population with a YFP+CFP fluorescence phenotype. Error bars = SD of technical qPCR triplicates. *r* is the Pearson correlation coefficient and *P* its p-value. RFU = relative fluorescence units, line = linear fit.

The following source data and figure supplements are available for figure 1:

**Source data 1.** Chromosomal coordinates of reporter cassette insertion loci.
**Source data 2.** Source data for *Figure 1E*.

*Figure 1 continued*

**Figure supplement 1.** Fine-scale determination of MICs of tetracycline for ancestor strains used in experimental evolution.
**Figure supplement 1—source data 1.** Triplicate $OD_{600}$ values (plate reader units) across tetracycline concentrations.

equal for all strains. In this way, the probability of rescue (estimated by performing a large number of replicate rescue experiments) is expected to be informative about the strain-specific rate and fitness effect of adaptive mutations of all types. Specifically, we adjusted the tetracycline concentrations used in evolution experiments to strain-specific minimum inhibitory concentrations (MICs), which we measured precisely (*Figure 1—figure supplement 1*). Given the otherwise isogenic background of the strains, we interpret MICs as a proxy for initial expression of *tetA-yfp*. MIC measurements revealed locus-dependent differences in the initial sensitivity to tetracycline, and all strains showed an increased MIC compared to the cassette-free ancestor, which indicates low baseline expression of *tetA-yfp*. For evolution experiments, we used tetracycline concentrations starting at 50% of the strain-specific MICs (*Figure 1D*).

We evolved 95 populations of each strain and measured optical density ($OD_{600}$) and fluorescence daily. Populations yielding $OD_{600}$ above a fixed threshold after ten days were regarded as rescued. Rescued populations were assigned to fluorescence phenotypes (YFP or YFP+CFP) based on the increase in fluorescence at the end of the experiment compared to the ancestor (*Figure 1C*). We performed qPCR on genomic DNA of populations displaying increased CFP fluorescence and found a good correlation between CFP fluorescence and the chromosomal copy number of the *tetA-yfp* gene (*Figure 1E*). Thus, CFP fluorescence is a valid proxy for the extent of high level amplification of the reporter cassette.

## The chromosomal location of a selected gene has large effects on adaptation

The number of rescued populations differed significantly between strains (*Figure 2A*), showing that the chromosomal location of the *tetA-yfp* gene is critical for its adaptive potential. No rescue was observed without the reporter cassette (*Figure 2—figure supplement 1*), and all rescued populations displayed increased YFP fluorescence, suggesting that rescue depended on the presence and overexpression of *tetA-yfp*. To test if increased expression of *tetA-yfp* was indeed causative for rescue, we deleted the reporter cassette genes in single clones isolated from three different rescued populations. Deletions eliminated growth on tetracycline in all three cases (*Figure 2B*). A minority of populations went extinct despite transiently increased YFP fluorescence (37/290 extinct populations), illustrating how our experimental selection filters for mutations that increase *tetA-yfp* expression above a minimum level. Two sets of replicate experiments yielded qualitatively similar results (*Figure 2C*), although the number of rescued populations fluctuated considerably between replicates, which likely reflects both technical variability (for example, in the precise amount of transferred inoculum from day to day) as well as the inherent stochasticity of evolutionary rescue processes. Time-trajectories of $OD_{600}$ and OD-normalized YFP and CFP fluorescence of all evolved populations are available in *Supplementary file 1* and fluorescence phenotype classifications in *Supplementary file 2*.

## Amplification mediated by flanking homology is a main determinant of neighborhood-dependent adaptation

We next set out to identify which mutation types were responsible for locus-dependent differences in the number of rescued populations. Strain *B* gave the highest number of rescued populations, and 76/77 rescued populations of this strain had reporter cassette amplifications (*Figure 2A*). Rescue by amplification in the other three strains was rare (*Figure 2AC*), implying that large differences between strains were related to locus-specific amplification. According to the 'canonical' model, formation of amplifications is limited by the rate at which initial duplications are generated

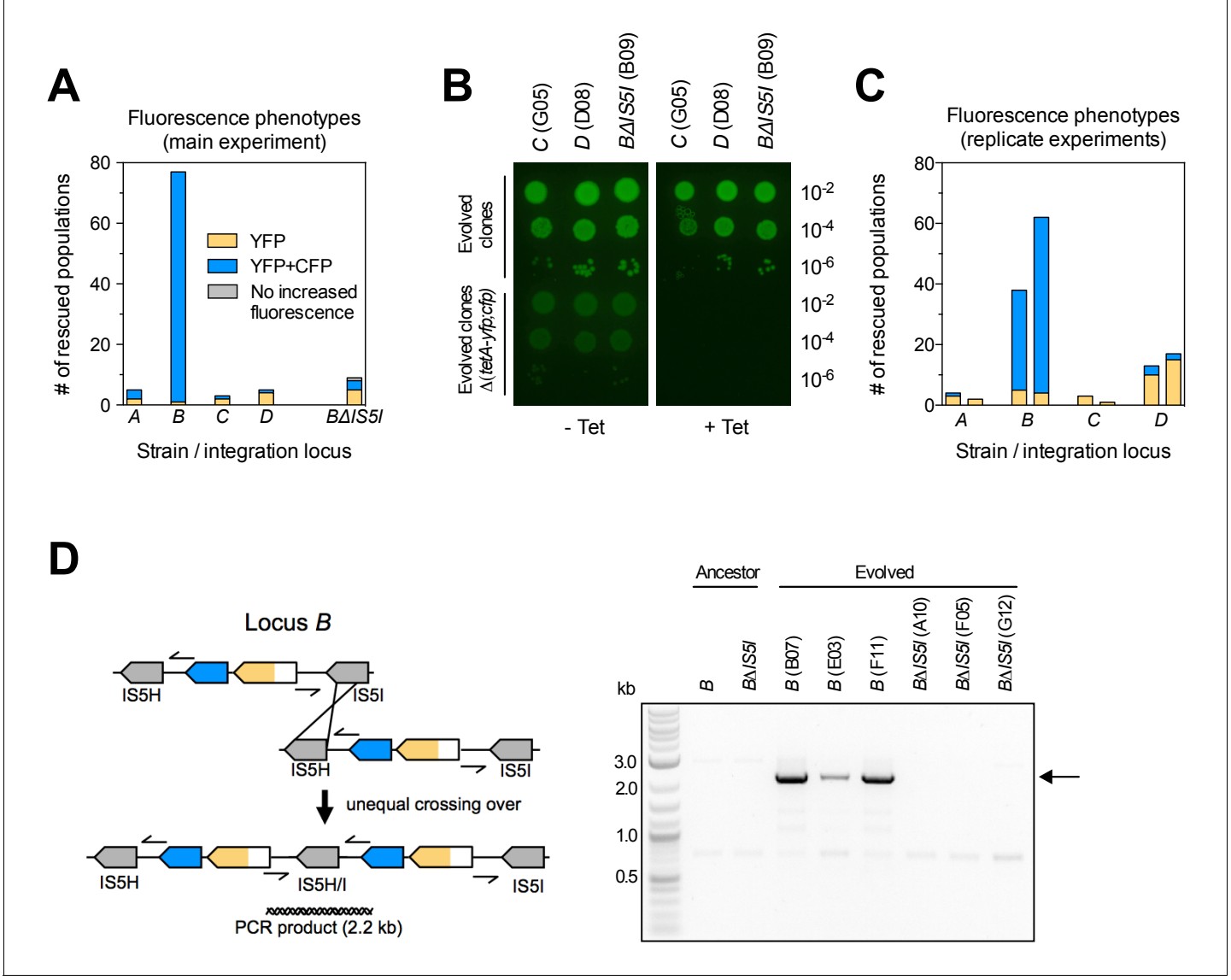

**Figure 2.** Large differences in adaptation by amplification depend on flanking homology in the chromosomal neighborhood. (A) Numbers of rescued populations by fluorescence phenotype. The numbers of rescued vs. extinct populations and the distribution of fluorescence phenotypes (YFP or YFP +CFP) differ among strains A, B, C, and D ($p<10^{-16}$ and $p<10^{-7}$, Fisher's exact test). (B) The ability of evolved clones to grow on tetracycline depends on the reporter cassette. Pictures show YFP-fluorescence of cultures spotted at different dilutions on solid medium with and without tetracycline (2.25 μg/mL). Top rows: evolved clones sampled from rescued populations of three different strains. Bottom rows: respective deletion mutants lacking reporter cassette genes. In parentheses: position of the sampled populations on 96-well plates in evolution experiments. (C) Numbers of rescued populations by fluorescence phenotype in two additional replicate sets of evolution experiments. (D) IS5 copies flanking locus B promote duplication. Left: Cartoon showing the position of the reporter cassette between two copies of IS5 (distances not drawn to scale, genes in between omitted) and the putative unequal crossing over-event causing initial duplications. Right: The expected amplicon junction is present in amplifications in strain B, but not in the ancestor or in amplifications in strain BΔIS5I. Arrow: junction PCR product obtained with outward facing primers shown as pointers in the cartoon on the left.

The following source data and figure supplements are available for figure 2:

**Figure supplement 1.** Survival curves of 95 populations in evolution experiments.

**Figure supplement 1—source data 1.** Numbers of populations with $OD_{600}>OD_t$.

**Figure supplement 2.** PCR products confirming the deletion of reporter cassette genes in clones shown in *Figure 2B*.

(*Romero and Palacios, 1997*). Rates of spontaneous duplication are elevated between homologous sequences such as rRNA operons or duplicate copies of insertion sequences (IS) due to frequent unequal crossing-over (*Anderson and Roth, 1981*; *Andersson and Hughes, 2009*). We found homologous copies of IS5 at either side of locus *B* (IS5H and IS5I), but no flanking homology in the chromosomal neighborhood of the other three loci. We verified the presence of IS5 at the boundary of the amplicon in rescued *B* populations by obtaining a PCR product of the expected junction in 16/16 tested clones of evolved populations (PCR products of three populations shown in *Figure 2D*). The junction was undetectable in the ancestor. Deleting one of the two flanking IS (strain *BΔIS5I*) gave highly reduced numbers of rescued populations (*Figure 2A*) and only a minority (3/9) had increased CFP fluorescence, which was not connected to amplification between the IS5H and IS5I (*Figure 2D*). These results confirm flanking homology and its effect on gene amplification as a main factor of chromosomal neighborhood on adaptation by increased gene expression.

## Adaptation involves a broad diversity of mutation types

Given the above result, we expected differences to disappear in the absence of IS and we repeated the evolution experiments with four strains that had the reporter cassette integrated at the same four loci as before, but that are derived from a multiple deletion strain (MDS42) free of all IS elements (*Pósfai et al., 2006*). MDS42 lacks around 15% of the MG1655 chromosome, including all prophages and many nonessential genes. Apart from the absence of IS-related mutations, the rates of other mutation types in MDS42 are similar to those in MG1655 (*Pósfai et al., 2006*). Loci *A-D* are not immediately next to genes absent in MDS42, the chromosomal neighborhood at a larger scale, however, is different between IS-wt and IS-free versions of the strains (*Figure 3—figure supplement 1*).

Despite the expected absence of frequent amplification of locus *B* in the IS-free genetic background, the fraction of rescued populations was still different among strains (p=3 × 10$^{-5}$, Fisher's exact test), and rescue was observed only in strains *B* and *D* (10 and 8 rescued populations, respectively). To explain these remaining differences, we identified candidate rescue mutations in strains with and without IS. Sequencing ~1 kb of DNA upstream of *tetA-yfp* revealed mutations of different types: point mutations (including small insertions and deletions), larger deletions, and insertions of mobile elements (*Figure 3AB* and *Figure 3—figure supplement 2*). The relative contribution of the different mutation types to adaptation differed between different chromosomal loci in both IS-containing and IS-free strains (p=10$^{-9}$ and p=0.003, Fisher's exact test). In several cases, mutations co-occurred with other mutations or amplifications (colored dots in *Figure 3A*), suggesting interactions between mutations, some of which we explored in more depth later (section 'Chromosomal neighborhood influences adaptation by affecting the fitness cost of amplifications').

We then continued to identify the mutation types responsible for the remaining differences in adaptation among strains, independent of neighborhood-dependent amplifications as described above. In order to test the effect of mutations on downstream expression independent of chromosomal locus, we constructed *yfp* reporter plasmids with all mutations found within the $p_0$ region of clones from rescued populations of the first replicate set of evolution experiments (IS-wt strains, IS-free strains and strain *BΔIS5I*, *Figure 3CDE*). Five of six small mutations altering the sequence of $p_0$ increased *yfp* fluorescence in plasmid reconstructions (*Figure 3C*), presumably by increasing the affinity of RNA polymerase to $p_0$. One mutation (T-145C) did not affect fluorescence and likely did not contribute to adaptation. Instead, rescue of the respective population, which also displayed a YFP+CFP fluorescence phenotype, likely depended on amplification alone. In contrast, two other point mutations identified in conjunction with amplifications (C-31T and G-92T), did increase reporter fluorescence on plasmids, providing examples of a combined beneficial effect of amplifications and additional mutations. Two of four insertions sequences that we had found inserted into $p_0$ increased reporter fluorescence on plasmids greatly (IS2 and IS3, *Figure 3D*), which is consistent with the delivery of outward-facing promoters within the termini of IS (*Mahillon and Chandler, 1998*). The two other IS (IS1 and IS5) had no or no strong effect on plasmid reporter fluorescence. Since some IS have been reported to contain partial outward-facing promoters that can drive downstream expression after insertion next to a resident complementary partial promoter site (*Mahillon and Chandler, 1998*), we tested IS1 and IS5 in the precise sequence context of $p_0$ in which these IS were found in evolution experiments (*Figure 3E*). In this sequence context, IS1 indeed increased reporter fluorescence, which depended on the 20 bp downstream of the insertion point

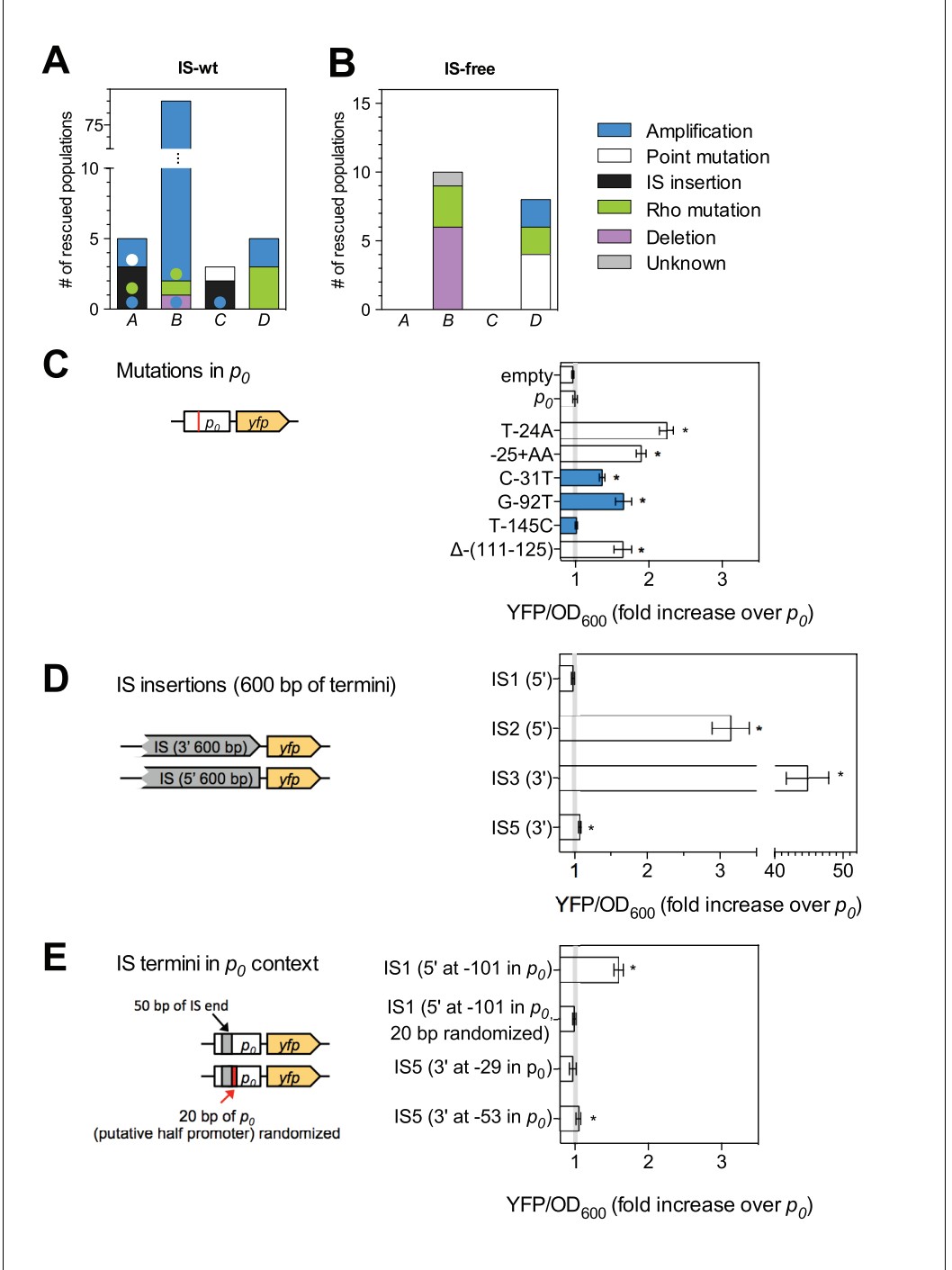

**Figure 3.** Adaptation involves a broad diversity of mutation types. Mutation types in rescued populations of IS-wt (**A**) and IS-free (**B**) strains. Colored dots = later mutations occurring on top of other mutations (see Methods). Mutation types differ between loci ($p=10^{-9}$ (**A**) and $p=0.003$ (**B**), Fisher's exact test). (**C–E**) Effect of reconstructed point mutations and IS insertions on reporter expression on plasmids. Plasmids contain mutations reconstructed upstream of a ribosomal binding site (not shown) and a *yfp* reporter gene as shown in cartoons. Empty = auto-fluorescence control (plasmid backbone); $p_0$ = ancestral 188 bp random sequence. Error bars = 95% confidence intervals of six technical replicates. Grey shading: 95% confidence interval of $OD_{600}$-normalized $p_0$ fluorescence. Asterisks = $p<0.05$, two-tailed t-test on mean fluorescence difference in comparison with $p_0$. (**C**) Reporter fluorescence driven by small mutations within $p_0$ (single bp substitutions and small insertions or deletions). Mutation coordinates = distance of mutation to start codon of *yfp*. Blue bars = mutations that co-occur with amplifications and show overlapping peaks in the sequence chromatogram of evolved clones, indicating presence

*Figure 3 continued on next page*

*Figure 3 continued*

of mutations only in a subset of copies in an amplification. (**D**) Reporter fluorescence driven by IS insertions. Plasmids contain the termini of IS which were truncated to 600 bp. 5' and 3' refers to the direction of the IS-contained transposase gene. IS2 and IS3 drive strong fluorescence of *yfp* in the plasmid context; IS1 and IS5 do not. (**E**) Reporter fluorescence driven by IS in the precise sequence context of $p_0$. IS1, but not IS5, contains a partial promoter whose activity depends on the adjacent sequence in $p_0$. Numbers in parentheses = distance between insertion point and the *yfp* start codon. 'rnd' = random shuffling of 20 bp of $p_0$ downstream of the IS1 insertion point.

The following source data and figure supplements are available for figure 3:

**Source data 1.** Source data for *Figure 3C–E*.

**Figure supplement 1.** Differences in the chromosomal neighborhood (100 kb) of loci *A-D* between IS-wt and IS-free strains.

**Figure supplement 2.** Graphical overview of mutations identified by sequencing.

**Figure supplement 3.** An upstream IS5 insertion in the chromosomal context of the reporter cassette confers resistance to tetracycline and increases *tetA-yfp* fluorescence.

**Figure supplement 4.** Mutations identified by whole genome sequencing of clones from four rescued populations of IS-wt strain *D*.

**Figure supplement 5.** Numbers of rescued populations by mutation type in two additional replicate sets of evolution experiments.

within $p_0$ (*Figure 3E*), consistent with the delivery of a half-promoter within the terminus of this IS. Insertion of IS5, which we repeatedly observed in evolution experiments, had very weak, but significant effects on downstream fluorescence on plasmids (*Figure 1DE*). To confirm the adaptive role of upstream IS5 insertions in the evolution experiments, we transduced one of the observed upstream IS5 insertions into the ancestral background, which restored growth on tetracycline as well as a marked increase in YFP fluoresence (*Figure 3—figure supplement 3*). Thus, in the chromosomal context, IS5 does increase expression of downstream genes, possibly due to effects on DNA bending (*Zhang and Saier, 2009*), which may not be recapitulated on the plasmid reconstruction. These results illustrate the diverse ways in which IS can adaptively affect gene expression, both dependent (IS1, IS5) and independent (IS2, IS3) of the insertion context. Given the reporter plasmid results and the fact that the same $p_0$ sequence is part of the reporter cassette at all four chromosomal loci, point mutations and IS insertions likely were not responsible for the observed differences in the frequency of rescue between strains that are not explained by amplifications.

## Properties of upstream genes determine the availability of two different types of adaptive promoter co-option mutations

Whole genome sequencing of clones from three rescued populations with neither upstream genetic changes nor amplifications (*Figure 3—figure supplement 4*), as well as subsequent screening of other rescued populations, revealed another candidate type of adaptive mutations, which altered the protein sequence of *rho* (*Figure 4—video 1*). Unlike mutations of the other types, *rho* mutations occurred in trans with respect to the reporter cassette. The *rho* gene of *E. coli* is an essential gene that encodes a transcriptional termination factor estimated to be required for termination at around half of all termination sites in *E. coli* (*Ciampi, 2006*). Contrary to point mutations and IS insertions, which we found upstream of all four loci (*Figure 3A* and *Figure 3—figure supplement 5*), Rho mutations and also upstream deletions were only found in evolved clones of strains with the reporter cassette at locus *B* or *D*, with one exception of a Rho mutation co-occurring with an upstream IS insertion in strain *A*. Thus, upstream deletions and Rho mutations provide candidates for locus-dependent adaptive mutations. Comparing the upstream neighborhood of the four different loci revealed the basis of this locus-dependency (*Figure 4A* and *Figure 4—figure supplement 1*). The

orientation and expression of upstream transcripts as determined in a different study (*Conway et al., 2014*) suggests that in strains *B* and *D*, active upstream promoters were co-opted to *tetA-yfp*, either by deletion of intervening genes, or by compromising Rho-dependent termination by partial-loss-of-function mutations in Rho that cause transcriptional read-through into *tetA-yfp*. At loci *A* and *C*, such adaptive mutations were not available because of two kinds of constraints from neighboring genes: either intervening genes were essential (constraining adaptive deletions, *Figure 4A*), or no upstream Rho-terminated transcripts were present (constraining adaptive Rho mutations, *Figure 4A*).

Since active transcripts shown in *Figure 4A* were experimentally determined under conditions different from our evolution experiments (*Conway et al., 2014*), and classification of termination sites as intrinsic or Rho-dependent was done only computationally (*Kingsford et al., 2007*; *Conway et al., 2014*), we experimentally assessed the effect of Rho mutations on transcriptional read-through across candidate upstream terminators at all four loci under experimental conditions approximating those in evolution experiments. We first confirmed the neighborhood-dependent effect of two different Rho mutations (S153F and M416I) on the phenotype of interest, that is, tetracycline resistance, by transduction into the ancestral IS-wt strains, which are isogenic except for the position of the reporter cassette (*Figure 4B*). Consistent with the presence of upstream Rho-terminated transcripts as shown in *Figure 4A*, an increased tolerance of Rho-mutants to tetracycline was observed only in strains with the reporter cassette at loci *B* and *D*, matching our observation that Rho-mutants were only found in rescued populations of these strains. We then performed PCR on cDNA prepared from a Rho-wt strain and a Rho mutant (M416I) strain grown in sub-inhibitory tetracycline (*Figure 4C*). We obtained PCR products consistent with read-through across candidate terminators upstream of locus *B* (downstream of *yeeD*) and locus *D* (*mnmG*), but not upstream of locus *A* (*cysS*) and locus *C* (*xapR*). A read-through transcript at locus *D* was detectable even in the Rho-wt background, which offers an explanation for the higher initial TetA-YFP expression observed in strain *D* (*Figure 1—figure supplement 1*). Mutations found in rescued populations of additional replicate experiments (fluorescence phenotypes in *Figure 2B*) are consistent with the above constraints on promoter co-opting mutations (*Figure 3—figure supplement 5*). Thus, upstream deletions and *trans* mutations that compromise transcriptional termination are mutation types that depend on the chromosomal neighborhood of the gene under selection. Specifically, the orientation, expression, essentiality, and termination mode of neighboring genes shape the fitness effect of these promoter co-option mutations.

## Chromosomal neighborhood influences adaptation by affecting the fitness cost of amplifications

As seen from promoter co-opting mutations, chromosomal neighborhood may affect the adaptive potential of a gene by influencing not only mutation rates (as flanking homology does for duplications that can expand into amplifications), but also mutation fitness effects. We next asked if this applies to amplifications as well. Due to the instability of amplifications and related difficulties in detecting them, quantifying the fitness effect of amplifications is laborious (*Adler et al., 2014*) and has so far not been done on a genome-wide scale. The benefit of amplifying a selected gene is counteracted by a cost that arises in part due to dosage imbalances in the co-amplified neighboring genes. This cost limits the ability of amplifications to effectively expand at the population level as selection increases, an ability that comes from high rates of expansion of amplifications at the level of the individual chromosome by homologous recombination. The probability of an amplification to contain a costly gene is expected to increase with the length of the amplicon.

We used two-color fluorescence data to extract information on amplification cost and its effect on adaptation. For validating this approach, we used two strains (strain *BΔIS5I*, and the newly created strain *E*, *Figure 5A*) that we predicted to form amplifications of higher and lower cost, respectively, when compared to the IS-containing strain *B*, which serves as reference. The IS5 deletion in strain *BΔIS5I*, which reduces the rate of duplications that kick-start amplifications (see above), is also expected to increase the fitness cost of amplifications, since amplicons may be larger than the 35 kb between IS5I and IS5H. In strain *E*, we placed the reporter cassette between two copies of IS1, where duplications are expected to form frequently, and where amplifications, due to small amplicon size (11 kb), are expected to expand at low cost. In our experiment, if cost is negligible, amplifications are expected to expand continuously as tetracycline selection increases, resulting in rescue. In

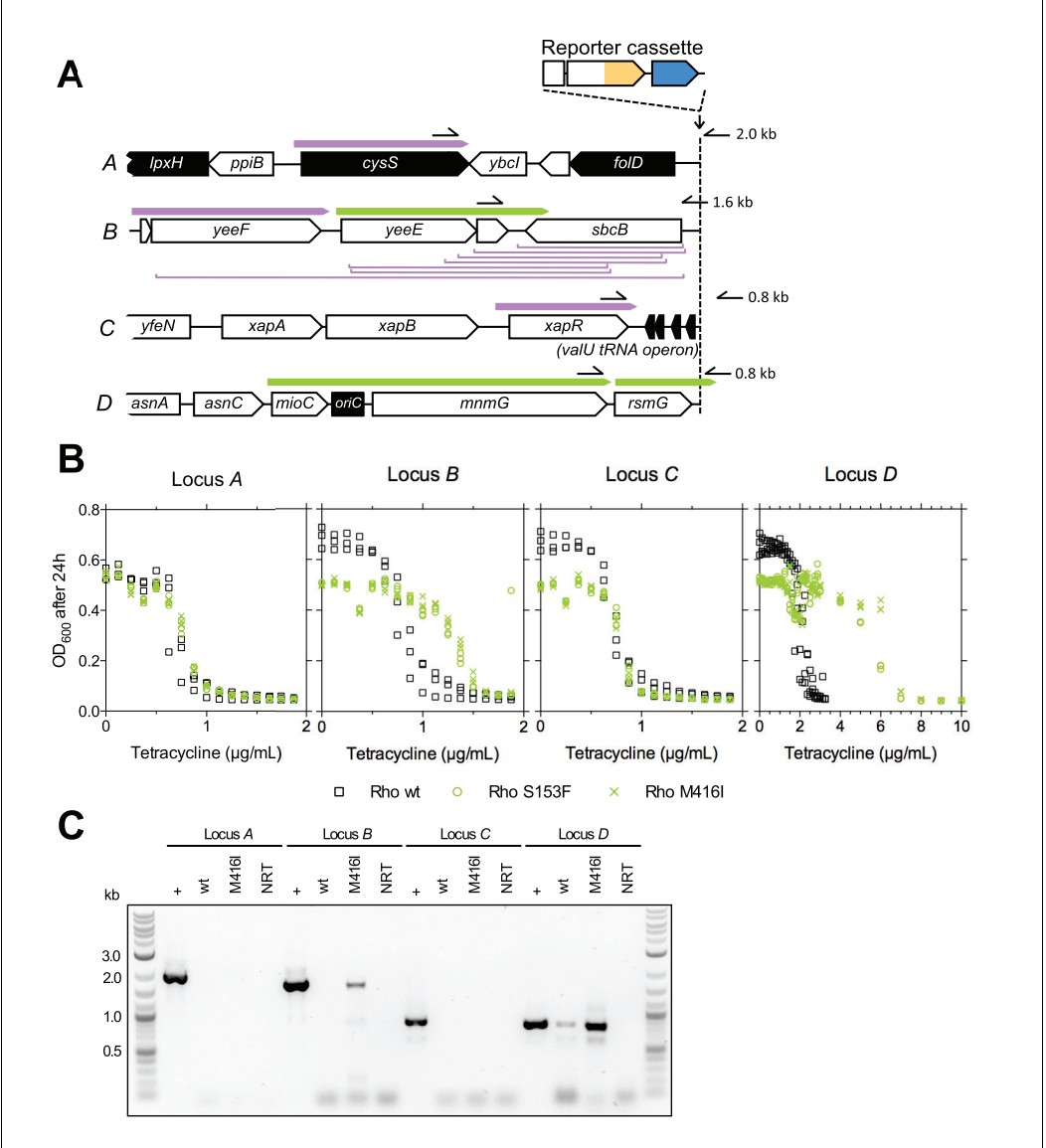

**Figure 4.** The fitness effect of promoter co-opting deletions and Rho-mutations depends on properties of upstream neighboring genes. (**A**) Genes and transcripts upstream of loci *A*, *B*, *C*, and *D*. Promoters of intrinsically terminated transcripts (purple) can be co-opted by deletions (purple brackets) if no essential gene (black arrows) is deleted. Promoters of Rho-terminated transcripts (green) can be co-opted by deletions or by partial loss-of-function mutations in Rho. Only putatively expressed transcripts oriented toward the reporter cassette are shown (all transcripts in *Figure 4—figure supplement 1*). Pointers and numbers on the right = position and size of PCR products shown in (**C**). (**B**) Tetracycline dose-response curves of strains with wt (black squares) or transduced mutant Rho (green circles = S153F, green crosses = M416I). Final $OD_{600}$ after 24 hr (platereader units) was measured in three biological replicates. Rho mutants are more tolerant to tetracycline only with the reporter cassette at loci *B* and *D*. (**C**) Read-through transcripts spanning upstream terminators in a Rho-mutant background are detectable at loci *B* and *D*, but not at loci *A* and *C*. Bands show PCR products obtained from genomic DNA (+ control) or cDNA from a Rho-wt or Rho mutant (M416I) strain grown with sub-inhibitory levels of tetracycline. Positions of used primers as indicated in (**A**). NRT = negative control (no reverse transcriptase).

The following source data and figure supplement are available for figure 4:

**Source data 1.** Source data for *Figure 4B*.

**Figure supplement 1.** Fully annotated genes and putatively expressed transcripts of either orientation upstream of the reporter cassette insertion loci.

**Figure 4—video 1.** Animated structure of the Rho hexamer with mutated residues highlighted. Mutations were mapped on the previously published structure of Rho (*Skordalakes and Berger, 2003*).

*Figure 4 continued on next page*

*Figure 4 continued*

**Figure 4—video 1—source data 1.** Rho mutations from all replicate evolution experiments.

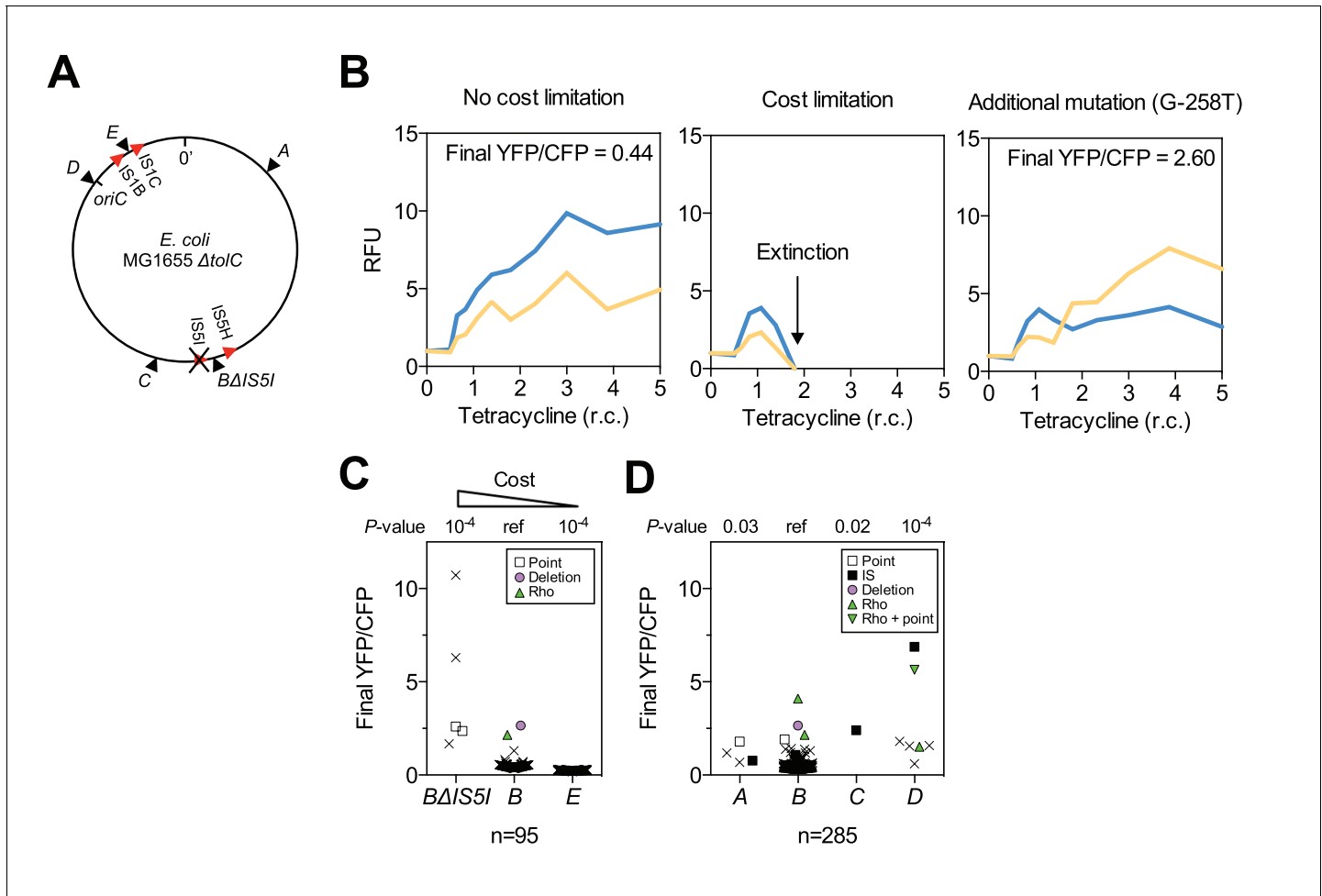

**Figure 5.** Chromosomal neighborhood influences the fitness cost of amplifications. (A) Chromosomal location of reporter cassette in strains *BΔIS5I* and *E* (IS distances not drawn to scale). (B) Example fluorescence trajectories. Left: low-cost amplifications expand in correlation with the increase in tetracycline concentration over 10 days. Middle: Cost-limited amplifications fail to expand at higher tetracycline concentrations resulting in extinction. Right: Amplifications can escape extinction in combination with other mutations increasing *tetA-yfp* expression, resulting in higher final YFP/CFP. RFU = relative fluorescence units (see Materials and methods), r.c. = relative concentration as multiples of MIC. (C) Final YFP/CFP ratios of rescued amplifications in strains expected to have a higher (strain *BΔIS5I*) or lower (strain *E*) cost of amplifications compared to strain *B*. n = initial number of replicate populations used for analysis. Crosses = populations rescued by amplifications without additional mutations. Other symbols = secondary mutations (see legend). p-values: permutation tests in comparison with strain *B*. (D) Final YFP/CFP ratios of rescued amplifications in strains *A-D*. n = 285 includes replicate evolution experiments to increase statistical power. Symbols and p-values as in (C).

The following source data and figure supplement are available for figure 5:

**Source data 1.** Source data for *Figure 5C and D*.

**Figure supplement 1.** Rescued populations of strains *BΔIS5I* and *E* by mutation type.

this case, YFP+CFP fluorescence increases in correlation with the level of tetracycline selection (*Figure 5B*, left). If amplifications are cost-limited, two outcomes are possible: (i) amplifications fail to expand beyond a certain level below that required for rescue, resulting in extinction – if the level of expansion before extinction is high enough, it will appear as a transient increase of CFP fluorescence in the fluorescence trajectories of extinct populations (*Figure 5B*, middle), (ii) amplifications allow rescue through interaction with other adaptive mutations that increase *tetA-yfp* expression – resulting in increased YFP/CFP fluorescence ratios in rescued populations (*Figure 5B*, right). We compared the numbers of extinct vs. rescued populations with (transiently) increased CFP fluorescence and found, as expected, that amplifications in strain *BΔIS5I* had a significantly higher extinction risk than amplifications in the reference strain *B* (*Table 1*, upper part), and rescued amplifications had significantly higher final YFP/CFP ratios (*Figure 5C*), confirming that low numbers of rescue in strain *BΔIS5I* were in part due to amplification costs. Contrariwise, populations with increased CFP fluorescence in strain *E* never went extinct (*Table 1*, upper part) and had consistently low final YFP/CFP ratios (*Figure 5C*), indicating the absence of a cost limitation.

Having validated extinction risk (*Table 1*) and final YFP/CFP ratios (*Figure 5C*) as indicators of amplification cost, we tested if neighborhood-dependent amplification costs had affected adaptation in strains *A*, *C*, and *D*. A significantly elevated extinction risk of populations with amplifications in strain *A* (*Table 1*, lower part), and significantly elevated final YFP/CFP ratios in connection with diverse additional mutations in strains *A*, *C*, and *D* (*Figure 5D*), support that the costs of amplifications in these strains were higher compared to strain *B*. Thus, amplification costs represent another neighborhood-dependent constraint on adaptation. In this perspective, the availability of neighborhood-dependent promoter co-option mutations, the most prevalent non-amplification mutation types at loci *B* and *D*, is an important determinant of adaptive potential not only in itself, but also in the interaction with amplifications.

## Chromosome neighborhood effects on adaptation in a single-step plating experiment

We next investigated whether our observations of chromosomal neighborhood effects on adaptation transfer to different selective conditions. In particular, we tested the possibility that differences in rescue between strains were due to different population sizes and thus different chances for beneficial mutations to occur, rather than due to different mutation rates or fitness effects as we propose. Our experimental design corrects for population size differences between strains at the first day of

**Table 1.** Differences in amplification cost indicated by the extinction risk of populations with amplifications. Populations with amplifications of higher (strain *BΔIS5I*) or lower (strain *E*) expected cost of amplifications have a higher or lower risk of becoming extinct, respectively. n = initial number of replicate populations used for analysis (n = 285 includes replicate evolution experiments to increase statistical power), 'Extinct' and 'Rescued' = numbers of extinct and rescued populations with amplifications as indicated by (transiently) increased CFP fluorescence (see Materials and methods), sample odds ratio compared to strain *B*, p-values: 2 × 2 Fisher's exact test.

| n | Strain | Populations with (transiently) increased CFP fluorescence | | Sample Odds Ratio | P-value |
| | | Extinct | Rescued | | |
|---|---|---|---|---|---|
| 95 | *BΔIS5I* | 12 | 5 | 10.1 | $10^{-4}$ |
| 95 | *B* | 18 | 76 | 1 (ref) | – |
| 95 | *E* | 0 | 95 | 0 | $10^{-6}$ |
| | | | | | |
| 285 | *A* | 8 | 4 | 5.8 | $10^{-3}$ |
| 285 | *B* | 58 | 168 | 1 (ref) | – |
| 285 | *C* | 0 | 1 | 0 | n.s. |
| 285 | *D* | 0 | 7 | 0 | n.s. |

selection (*Figure 1—figure supplement 1*, and first section of the results part), but not necessarily for population size differences at later days of the experiments. Therefore, we performed single-step plating experiments, in which approximately the same numbers of cells are plated for every strain. In these Luria-Delbrück-type experiments, we plated replicate cultures grown under non-selective conditions on solid medium with tetracycline at two-fold MIC levels. We scored the number of colonies on each plate after 2 days, when clearly visible colonies first appeared. These early colonies are expected to result mostly from pre-plating single-step mutations that increase *tetA-yfp* expression (point mutations, IS insertions, and promoter co-option mutations). As in evolution experiments, colony numbers in strains *B* and *D* were higher than in strains *A* and *C* (*Figure 6*, left), for both IS-wt and IS-free genetic backgrounds. This result is consistent with neighborhood-dependent availability of promoter co-option mutations as observed also in evolution experiments. High CFP fluorescence, indicative of amplifications, was observed only in a small fraction of early colonies (34 of 1661 across all strains and plates). During longer incubation, the number of colonies on plates of IS-wt strain *B* increased steadily (*Figure 6—figure supplement 1*) and almost all of these later colonies (1229/ 1304 on ten plates) showed high CFP fluorescence. Since tetracycline is bacteriostatic rather than bactericidal, the appearance of these late colonies can be explained by a continuous process of reporter cassette amplification expansion and increasing growth rates after plating on selective medium, starting from frequent duplications that have a slight growth advantage over single-copy cells (*Andersson et al., 1998*). After 5 days, colony counts on plates were qualitatively similar to rescue frequencies in evolution experiments, with IS-wt strain *B* giving the highest number of colonies (*Figure 6*, right). In all other tested strains, late colonies appeared at much lower rates (*Figure 6— figure supplement 1*) and did not show high CFP fluorescence in most cases (*Figure 6*, right, and *Figure 6—figure supplement 2*), reflecting the minor role of amplification in strains that lack flanking homology in the chromosomal neighborhood of the selected gene. The consistency between liquid-culture evolutionary rescue experiments and plating experiments supports that strong effects of chromosomal neighborhood on the rate and fitness effect of adaptive mutations extend to different selective regimes.

## Discussion

Our results reveal a complex genetic basis of strong effects of chromosomal position on the adaptive potential of a specific gene (*Figure 7A*). By combining time-resolved fluorescence data from the reporter cassette and end-point genetic analysis, we demonstrate how the relative contribution of previously known mutation types to adaptation (*Figure 7A*, bottom row) differs between chromosomal loci, how these differences arise, and how a layer of complexity is added by the interaction of mutation types. Thus, the concept of a one-dimensional mutation rate and a focus on point mutations can be misleading (*Martinez and Baquero, 2000*), even for the simple case of adaptation by increased expression of a single gene. Instead, the adaptive potential of a given gene is a system-level property shaped by the local chromosomal genetic neighborhood. Consequently, the organization of genes on a chromosome is both cause and consequence of evolutionary change.

Importantly, the effects that we describe arise from several properties (*Figure 7A*, top row) of different genetic elements that are present in the vicinity of the selected gene, rather than from more global factors such as distance to the origin of replication or chromosome macro-domain organization (*Bryant et al., 2014*). Therefore, we propose to refer to them as 'chromosome neighborhood effects' that determine the evolution of gene expression, as opposed to 'chromosome position effects' that modulate gene expression per se (*Bryant et al., 2014*; *Levis et al., 1985*; *Akhtar et al., 2013*).

### Different mutation types interact to cause neighborhood-dependent differences in adaptive potential

In our experiments, chromosomal neighborhoods facilitate or constrain adaptation mainly via amplification and promoter co-option mutations, by affecting the rate of mutations (duplication-amplification) or the fitness effects of mutations (promoter co-option mutations and amplifications). For gene amplification, a strong effect of flanking homology as provided by IS, which are often present in multiple copies, has been known for a long time (*Peterson and Rownd, 1985*; *Andersson and Hughes, 2009*). Our data confirm that if flanking homology is present at a given locus, amplification is the

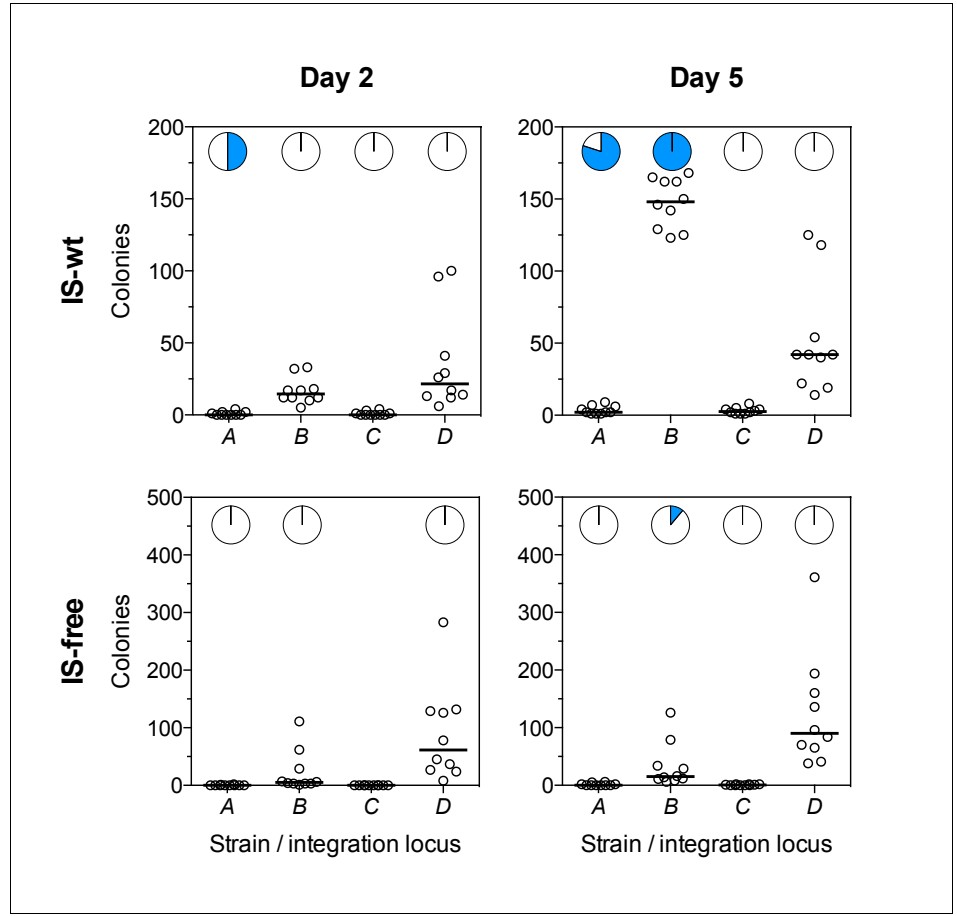

**Figure 6.** Tetracycline-resistant mutants arising in a single-step plating experiment. For each strain (top panels = IS-wt, bottom panels = IS-free), 10 replicate cultures grown in the absence of tetracycline were plated on agar with tetracycline concentration two times the strain-specific MIC. Left: Colony counts after 2 days of incubation. Right: Colony counts after 5 days of incubation. Horizontal lines show the median colony number from 10 replicate plates. Pie charts = fraction of plates in which a single tested colony appearing at day 2 (left) or at days 4–5 (right) showed high CFP fluorescence indicative of amplification (*Figure 6—figure supplement 2*).

The following source data and figure supplements are available for figure 6:

**Source data 1.** Numbers of colonies on replicate plates at days 2 and 5.

**Figure supplement 1.** Colony appearance over time in plating experiments.

**Figure supplement 1—source data 1.** Numbers of colonies on replicate plates (time-course).

**Figure supplement 2.** CFP-fluorescence of cultures spotted on non-selective medium used to obtain pie-chart data in *Figure 6*.

**Figure supplement 2—source data 1.** Mean fluorescence intensity values of culture spots and thresholding for identification of colonies with extensive amplifications.

---

main response to selection for increased gene expression. For loci lacking nearby flanking homology, which depending on the distribution of IS elements on a chromosome may be the majority of loci (*Boyd and Hartl, 1997*; *Green et al., 1984*), our data show that adaptation by amplification is limited on the level of duplication rate and fitness cost. For these loci, differences in the adaptive potential are largely due to the different availability of deletions and mutations compromising

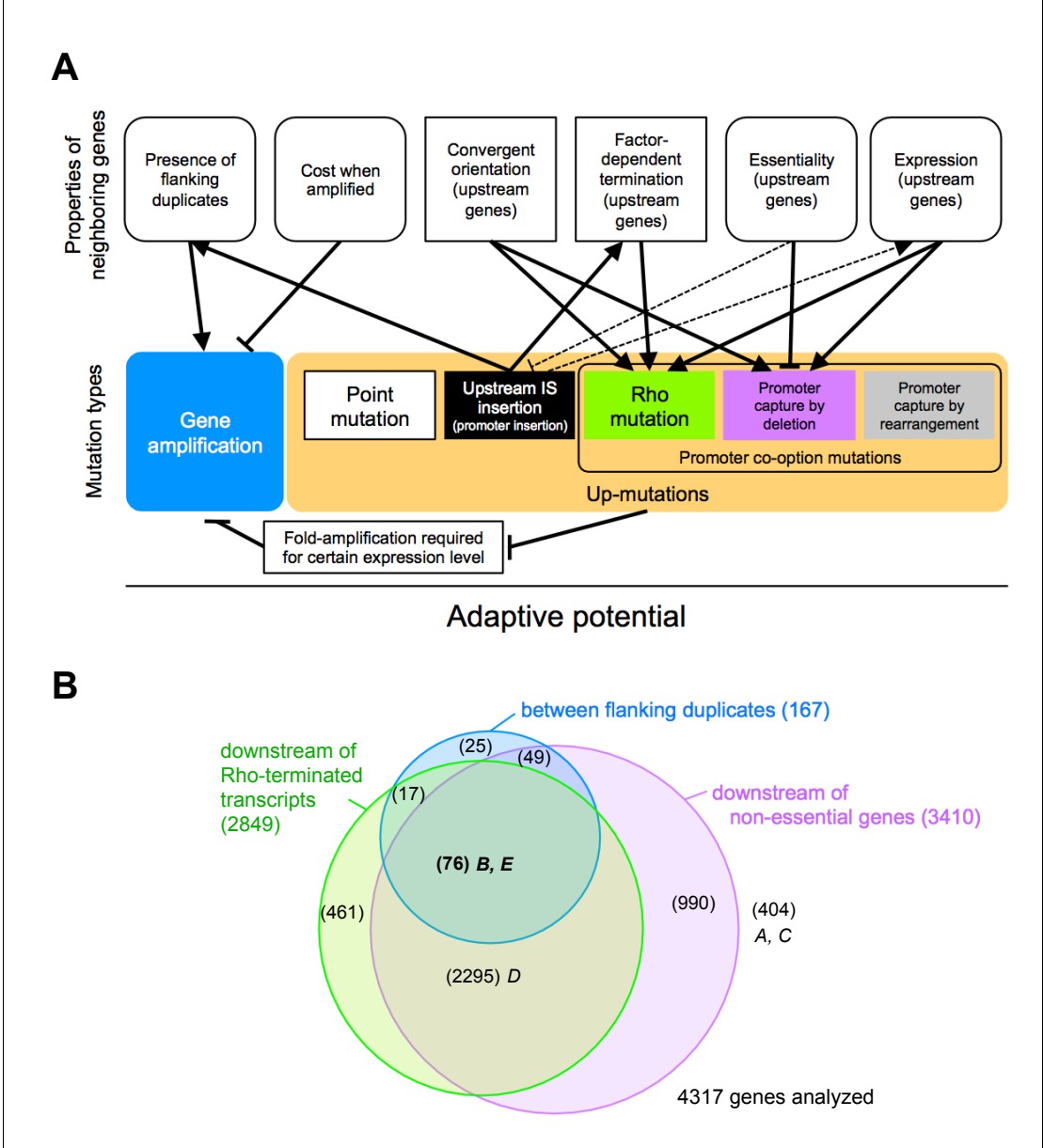

**Figure 7.** The adaptive potential of a gene under selection for increased gene expression as a complex function of properties of neighboring genes that affect and are affected by mutations of diverse types. (**A**) Top row: Properties of neighboring genes that we identify as main determinants of the adaptive potential of a gene given its chromosomal neighborhood. Round corners indicate 'dynamic' properties that may be environment-dependent or subject to change over short evolutionary timescales. Bottom row: Different mutation types causing increased expression of a gene. Solid arrows: Effects and interactions shown or suggested by data in this study. Dashed arrows: Other effects and interactions that are likely to exist. Pointed arrowheads indicate a positive effect, T-bar ends indicate a negative effect. A sentence equivalent of each arrow is given in *Figure 7—source data 1*. As a sum of the above interactions, the adaptive potential of a gene emerges as a system-property. (**B**) Classification of chromosomal neighborhoods of *E. coli* genes according to adaptive potential. The chromosomal neighborhood of 4317 genes of *E. coli* MG1655 was assessed using published information on the position of promoters and terminators (*Conway et al., 2014*) and gene essentiality (see Methods for details). Numbers in parentheses = genes belonging to respective sets or intersections of sets. Genes in the intersection of all three circles (boldface) are expected to have the highest adaptive potential based on their chromosomal neighborhood. Loci *A-E* of this study are placed in the respective areas of the diagram.

The following source data is available for figure 7:

*Figure 7 continued on next page*

*Figure 7 continued*

**Source data 1.** Extended legend of *Figure 7A* explaining each arrow and what loci are affected by respective interactions.
**Source data 2.** List of *E. coli* genes included in the analysis shown in *Figure 7B* and their assignment to the three sets shown by colored circles.

transcriptional termination, both of which co-opt upstream promoters to the selected gene. Such mutations also act in concert with amplifications and can alleviate amplification cost limitations by lowering the required fold-amplification to reach a certain level of expression of the selected gene (*Figure 7A*).

The multitude of mutations discovered in the termination factor Rho suggests that the function of this protein may be more 'tunable' than expected from it being an essential gene in *E. coli*. Our results may suggest that adaptation via *trans* mutations in Rho with potentially large pleiotropic effects is more likely than via local mutations that compromise upstream terminators in cis. Given that the sequence-dependence of Rho-dependent termination is poorly understood (*Ciampi, 2006*), there is no clear expectation of the nature and target size of mutations that would compromise Rho-dependent termination in cis. This makes it difficult to compare adaptation via mutations affecting Rho-dependent termination in cis versus *trans*. The adaptiveness of *trans* mutations in Rho despite their pleiotropic effects is supported by a previously characterized single amino-acid substitution in Rho, which was found to have large-scale effects on the *E. coli* transcriptome and to confer higher fitness in several environments (*Freddolino et al., 2012*). We found substitutions at 22 different amino acid residues mapping to various regions of the Rho protein structure (*Skordalakes and Berger, 2003*) (*Figure 4—video 1* and *Figure 4—video 1—source data 1*), which largely expands the number of Rho residues found mutated in evolution experiments (*Conrad et al., 2011*). This supports the idea that operons delimited by factor-dependent terminators may be rather fluid, providing a large source of variation for adaptation to changing environments. It remains to be seen whether different Rho alleles, by revealing 'hidden' transcriptional variation, serve as capacitors of adaptation (*Masel, 2013*) beyond laboratory evolution experiments.

## Assessing properties of neighboring genes to infer the adaptive potential of a gene under selection

For both amplifications and promoter co-opting mutations, the influence of the chromosomal neighborhood arises mechanistically from several simple properties of neighboring genes—their expression, orientation, transcriptional termination, essentiality, the presence or absence of flanking gene duplicates—and from the cost of neighboring gene co-amplification (*Figure 7A*, top row). If these properties are known at a genomic scale, inferring a chromosome-wide 'map of adaptive potential' becomes conceivable. An understanding of adaptive potential may help assess the risk of resistance evolution via overexpression of preexisting chromosomal genes (as opposed to via acquisition by horizontal transfer). Clearly, some properties of neighboring genes can be assessed on a genome-wide scale more easily (e.g. gene orientation) than others (e.g. gene essentiality or cost of genes when amplified). Once it becomes feasible to acquire data on all the main factors shaping adaptive potential, this data may improve efforts to predict specific adaptations.

As a first step toward this goal, we used published information on gene essentiality, and promoter and terminator locations (*Conway et al., 2014*) to assess how many of *E. coli* genes (strain MG1655) are expected to reside in a chromosomal neighborhood associated with high adaptive potential (*Figure 7B*). Based on the most simply assessable properties (colored circles in *Figure 7B*), the chromosomal neighborhood of most genes (2295/4317) is expected to have a medium adaptive potential, comparable to that of locus *D* from this study.

## Adaptive potential as a dynamic property

Importantly, some properties of chromosomal neighborhoods are dynamic (rounded boxes in *Figure 7A*)—gene essentiality (*Baba et al., 2006*) and expression can be environment-dependent, and transposition causes rapid turnover of mobile element positions (*Sawyer et al., 1987*;

*Wagner, 2006*). Therefore, the classification of chromosomal neighborhoods of genes according to adaptive potential as in *Figure 7B* needs to be understood as a snapshot in time reflecting particular conditions. Also, how adaptive potential translates into the actual likelihood of adaptation depends on population parameters and the precise selection scenario.

On evolutionary timescales, the dynamics of chromosomal neighborhood properties would rapidly degrade signals that neighborhood-dependent evolution leaves in genome sequences. Nevertheless, neighborhood-dependent evolution could offer mechanistic explanations for phenomena observed in genomic data such as operon organization (*Reams and Neidle, 2004*; *Lawrence and Roth, 1996*), reductive genome evolution by promoter capture-deletions as suggested previously (*Lind et al., 2015*), or the chromosomal position of horizontally transferred genes (*Touchon et al., 2009*). Since horizontally transferred genes carrying selective functions are often silenced after initial integration (*Navarre et al., 2006*; *Cardinale et al., 2008*), they depend on activating mutations to play out their benefit to the host and become stably maintained in the host chromosome. Thus, the evolutionary fate of horizontally transferred genes will be shaped by the new chromosomal neighborhood they find themselves in. For example, a drug resistance gene entering the genome at loci *B* or *D* via horizontal transfer will be more likely to enable survival of the host under drug selection, compared to insertion at loci *A* and *C*, both because of higher initial expression and the higher adaptive potential associated with these loci as described here. The common association of horizontally acquired genes with flanking mobile elements as in complex transposons and genomic islands (*Dobrindt et al., 2004*) may not only reflect the high transferability of such configurations, but also their high amplifiability, which may be of particular relevance for mis-expressed foreign genes.

### Chromosomal neighborhood effects beyond prokaryotes

Although our results reflect many specifics of prokaryote genome organization, the importance of promoter-capture mutations (*ar-Rushdi et al., 1983*), modulation of transcriptional read-through (*Grosso et al., 2015*) and gene amplification (*Cole et al., 1992*; *Gajduskova et al., 2007*) extends to cancer evolution and cases of rapid adaptation in higher organisms (*Devonshire and Field, 1991*). This implies that chromosomal neighborhood effects on evolution may be of wider significance and they could be investigated with similar reporter-based methods.

## Materials and methods

### Materials

Unless noted otherwise, we obtained chemicals from Sigma-Aldrich (St. Louis, Missouri) and enzymes from New England Biolabs (Ipswich, Massachusetts). Evolution experiments and phenotyping tests were done in in M9 medium supplemented with 2 mM $MgSO_4$, 0.1 mM $CaCl_2$, and 0.2% glucose and 0.2% casein hydrolysate as carbon sources (M9CG medium), unless noted otherwise. A list of oligonucleotides, strains, and plasmids is available in *Supplementary file 3*.

### Construction of the reporter cassette

The reporter cassette ($p_0$-RBS-*tetA-yfp*-$p_R$-*cfp*) was assembled on a plasmid using a combination of standard cloning techniques, ligation chain reaction (*Rouillard et al., 2004*), and fusion PCR. For the $p_0$ sequence upstream of *tetA-yfp*, we generated a random 188 bp nucleotide sequence matching the average GC content of *E. coli* (CCGGAAAGACGGGCTTCAAAGCAACCTGACCACGGTTGCG CGTCCGTATCAAGATCCTCTTAATAAGCCCCCGTCACTGTTGGTTGTAGAGCCCAGGACGGG TTGGCCAGATGTGCGACTATATCGCTTAGTGGCTCTTGGGCCGCGGTGCGTTACCTTGCAGGAA TTGAGGCCGTCCGTTAATTTCC). We synthesized the sequence from oligonucleotides in a ligation chain reaction. The *tetA* sequence was taken from strain TKC (*Sharan et al., 2009*), and the *yfp* gene from plasmid pZA21-*yfp* (*Lutz and Bujard, 1997*). At the fusion point, we placed a 3xGS linker peptide between the C-terminus of TetA and the N-terminus of YFP. Between $p_0$ and the start codon of *tetA-yfp* is a sequence containing a restriction site and a ribosomal binding site (GTCGA-CAGGAGGAATTCACC). We placed the $p_0$-*tetA-yfp* sequence on plasmid pAH81-FRT-*cfp* (*Haldimann and Wanner, 2001*), upstream of the chloramphenicol resistance gene and the terminator-flanked $p_R$-*cfp* gene. $p_R$ is a strong constitutive promoter originating from phage λ. We sequenced the full length of the reporter cassette on the resulting plasmid, pMS7. Replication of the

pMS7 plasmid depends on the Pir protein and the plasmid was propagated in a *pir*-containing version of strain DH5α.

## Strain construction

We moved the *ΔtolC::kan* allele from *E. coli* strain JW5503-1 into strain MG1655 using P1 transduction. For the IS-free genetic background, the same *ΔtolC::kan* allele was introduced into strain MDS42 (*Pósfai et al., 2006*) by recombineering (*Thomason et al., 2014*) with pKD13 (*Datsenko and Wanner, 2000*) as PCR template. *kanR* cassettes were removed using plasmid pCP20 (*Datsenko and Wanner, 2000*). We inserted the reporter cassette from plasmid pMS7 into the two *ΔtolC* strains by recombineering. Precise insertion points are given in *Figure 1—source data 1*. All reporter cassette genes point toward the terminus of replication. Recombinants were selected on LB agar with chloramphenicol (10 μg/mL). The chloramphenicol marker was subsequently removed (*Datsenko and Wanner, 2000*). We confirmed the presence of the full-length single copy insertion by PCR and verified the sequence of $p_0$-*tetA-yfp* by sequencing. The presence of functional $p_R$-*cfp* was confirmed by observing fluorescence. To obtain strain *BΔIS5I*, the *camR* cassette from pKD3 (*Datsenko and Wanner, 2000*) was recombineered into the IS5I element of strain *B*. Recombinants were selected with choloramphenicol (10 μg/mL) and confirmed by PCR. Deletion of the reporter cassette genes in evolved clones was done by recombineering the *kanR* cassette of pKD13 into the reporter cassette such that the coding regions of both *tetA-yfp* and *cfp* were disrupted. Deletions were confirmed by absence of fluorescence and PCR with flanking primers (*Figure 2—figure supplement 2*). For P1 transduction of *rho* mutations, we first transduced mutations S153F and M416I from rescued clones of populations of strain *D* into MG1655. As selective marker, we used a *kanR* cassette that we had inserted upstream of *rho* by recombineering. After sequence verification, we transduced *rho* mutations into IS-wt strains *A-D*.

## MIC measurements and dose-response curves

Strains were pre-grown for 16 hr in M9CG medium without tetracycline and transferred to 96-well plates (200 μL/well). From there, we pin-diluted cultures with a VP408 pin replicator (V and P Scientific, San Diego, California, dilution factor ~1:820, tested with fluorescein) into fresh medium with different concentrations of tetracycline, incubated plates for 24 hr at 37°C on a Titramax plateshaker (Heidolph, Schwabach, Germany, 900 rpm), shook plates for 20 s at 1200 rpm and measured $OD_{600}$ with a H1 platereader (Biotek, Vinooski, Vermont). For obtaining fine-scale MIC measurements we tested tetracycline concentrations at intervals of 0.125 μg/mL. We defined MIC as the lowest drug concentration that yielded $OD_{600} \leq 0.075$ (plate reader units) in three replicates performed on different days.

## Evolution experiments

All precultures and evolution experiments were performed in M9CG medium. We transferred an overnight culture of every strain into 95 wells of clear flat bottom 96-well plates (200 μL/well), from where we diluted cultures into medium with tetracycline using VP408. One well contained a growth medium control. As initial concentration of tetracycline, we used half of the strain-specific MIC. For 10 days, we pin-diluted cultures with VP408 every 24 hr into medium with geometrically increasing tetracycline concentrations such that at day 10 the concentration was 10 times the initial concentration (*Figure 1D*). During the experiment, the maximum number of generations was set by the daily dilution factor (~1:820) and was ~97. A fresh tetracycline stock solution was prepared from powdered tetracycline-HCl every day. All incubations were done at 900 rpm on a plate shaker at 37°C in the dark and plates were wrapped in plastic bags to mitigate evaporation. Replicate evolution experiments were performed with two additional 96-well plates for each of strains *A*, *B*, *C*, and *D* (IS-wt). Each 96-well plate was started from a culture inoculated with a different colony. At the end of experiments, we froze all rescued populations.

## $OD_{600}$ and fluorescence measurements

Every day during the evolution experiment, after using 24-hr-old cultures for inoculating fresh medium with a higher tetracycline concentrations using VP408, we shook the old plates for 20 s at 1200 rpm to

resuspend cells and measured $OD_{600}$ and reporter fluorescence with a H1 Platereader (Biotek, Vinooski, Vermont; excitation/emission: YFP 515/545 nm / gain 100; CFP 433/475 nm / gain 60).

## Data analysis

Populations were classified as rescued if $OD_{600}$ exceeded 0.075 (plate reader units) at the end of the experiment. Fluorescence values were normalized to $OD_{600}$ and set to zero if $OD_{600}$ fell below 0.075. As reference for calculating the fold-increase in fluorescence, we took the average OD-normalized fluorescence of 95 cultures of the respective ancestral strain, inoculated in the same way as described for the beginning of evolution experiments, and grown in 96-well plates for 24 hr without tetracycline. Rescued populations were classified as YFP or YFP+CFP if the observed fold increase in respective fluorescence over the ancestor was >2.77 at the end of the experiment. This threshold corresponds to the lowest observed increase in YFP fluorescence that was sufficient for rescue in the first set of replicate experiments (IS-wt strains *A*, *B*, *C*, and *D*). To identify populations that went extinct despite elevated YFP and/or CFP fluorescence we applied more stringent criteria, requiring increased fluorescence (fold increase >2.77) for at least 2 days at which OD was >0.3 (platereader units). These criteria were used to exclude extinct populations that were false positive for increased fluorescence due to low $OD_{600}$ values prior to extinction. Rescued populations that met the more stringent criteria for elevated CFP fluorescence, but that did not show elevated CFP fluorescence at the end of the experiment (final fold increase <2.77), were counted as amplifications for cost analysis (*Figure 5* and *Table 1*), but not for *Figure 2*. For calculating final YFP/CFP ratios of rescued amplifications, we used internal plate reader fluorescence units directly. A Matlab script used to perform the above analysis is available as a supplementary file along with the platereader raw data used as input for the script (*Source code 1*). Plots of fluorescence trajectories of every population can be found in *Supplementary file 1* and phenotype classifications in *Supplementary file 2*.

## Quantitative PCR for reporter cassette copy number determination

We inoculated samples of all rescued populations that we had chosen for sequencing from the first set of replicate experiments and that had a YFP+CFP fluorescence phenotype. We inoculated 2 mL M9CG with 10 µL of populations that were frozen at the end of the evolution experiment. The large inoculum was used to maintain amplification-related population diversity. We added the same amount of tetracycline as on the last day of evolution experiments to maintain amplifications. From all cultures that were turbid after overnight incubation, we isolated genomic DNA (gDNA). Ancestor gDNA was isolated from cultures without tetracycline. We performed qPCR using the GoTaq qPCR mastermix (Promega, Madison, Wisconsin) and a C1000 instrument (Bio-Rad, Hercules, California). Using dilution series of one of the gDNA extracts as template, we confirmed that all primer pairs had an amplification efficiency >90%. We quantified the copy number of *tetA* in each sample with the $\Delta\Delta$Cq method implemented in the instrument software (Bio-Rad), taking amplification efficiency into account. As reference, we used loci equidistant from the origin of replication and compared ratios of the measured and reference locus to the ratio of the same two loci in the ancestral DNA. qPCR was done in three technical replicates.

## Identification of flanking homology

We searched 400 kb around loci *A-D* for homologous sequences on either side using REPuter (*Kurtz et al., 2001*) with the following search criteria: forward repeats $\geq$ 200 bp, Hamming distance $\leq$5.

## DNA sequencing

We streaked all rescued populations of strains *A*, *C*, *D* (IS-wt), of strains *B* and *D* (IS-free), and of strain *B∆IS5I* for single colonies on LB agar. For IS-wt strain *B*, we analyzed one rescued population that had a YFP-only fluorescent phenotype, two YFP+CFP populations with unusual fluorescence trajectories and 11 randomly chosen populations from the remaining 74 YFP+CFP rescued populations, which had highly similar fluorescence trajectories. Colony-PCRs were performed on a single representative clone of each streak. We amplified at least 1.5 kb of the region upstream of the *tetA* start codon. The size of PCR products was checked for insertions or deletions on an agarose gel. Sequences were obtained using primer tetA_pseq1_f. If no PCR product was obtained, we performed

arbitrary PCR with primer tetA_pseq2_f and a random primer, arb1 or arb6, for upstream binding. We then did a second PCR with a nested primer tetA_arb2 and primer arb2 using the first PCR product as template, and sequenced DNA extracted from the largest distinct band on an agarose gel. The full-length sequence of the *rho* gene was amplified and sequenced with primers rho_seq_f and rho_seq_r. For additional replicate evolution experiments, we sequenced clones of all rescued populations with a YFP fluorescence phenotype and with a YFP+CFP fluorescence phenotype showing high final YFP/CFP ratios. In four cases, we identified the exact same mutation in clones isolated from two populations that had been in neighboring wells during evolution experiments. In order to ensure that a potential cross-contamination between these two wells did not influence results, we excluded one of each pair of such neighboring populations from all analyses.

## Junction PCR

Colony PCR for amplification junctions was performed with primers IS5I_flank_f and IS5H_flank_r on single colonies of 16/16 evolved populations of strain *B*. For the data shown in *Figure 2D*, we used gDNA previously isolated from populations for qPCR to ensure a comparable amount of PCR template in all reactions.

## Whole genome sequencing

We isolated gDNA from overnight cultures of single clones of four rescued *D* populations as well as of the ancestral *D* strain grown in LB. A whole genome library was prepared and sequenced by GATC biotech (Konstanz, Germany) on an Illumina sequencer (125 bp reads). Fastq files were analyzed with the breseq script (*Barrick et al., 2014*). We used the MG1655 genome (Genbank accession number U00096.3) as a reference for assembling the ancestral *D* genome, which then served as a reference for analyzing the genomes of the evolved clones. Fastq files are available at: 10.15479/AT:ISTA:65.

## Cloning of reporter plasmids

For building the reference plasmid pAnc, which reports on expression from the ancestral $p_0$ sequence, we exchanged the pLtetO-1 promoter and RBS of pZA21-*yfp* for the $p_0$-RBS sequence upstream of *tetA-yfp* in the reporter cassette. Using a Q5 site-directed mutagenesis kit (New England Biolabs) with pAnc as template, we reconstructed small mutations (substitutions and small insertions and deletions, *Figure 3C*). We did the same with the terminal 50 bp of IS1 (5' terminus) and IS5 (3' terminus), which we put instead of the 50 bp of $p_0$ in the exact position where insertions were found in the experiment (*Figure 3E*). To confirm the IS1-$p_0$ hybrid promoter, we exchanged 20 bp of $p_0$ downstream of the IS1 insertion point in the respective reporter plasmid. The 20 bp were replaced by a randomly shuffled sequence composed of the same nucleotides. For the other IS reporter plasmids (*Figure 3D*), we PCR-amplified the last 600 bp of IS and cloned them into the XhoI/EcoRI sites of pZA21-*yfp*. The orientation of the truncated IS corresponds to that found in sequenced clones. As autofluorescence control, we removed the YFP fragment between EcoRI and MfeI restriction sites of pZA21-*yfp* and obtained pZA21-empty by religation of compatible ends. All changes were sequence-verified. Cloning and reporter measurements were done in strain NEB 5 alpha (New England Biolabs).

## Quantifying YFP reporter fluorescence from plasmids

We grew six replicate overnight cultures of the reporter plasmid strains in LB Kanamycin (50 µg/mL) in a 96-well plate and diluted them into M9CG supplemented with Kanamycin using a VP407 pin replicator (approximate dilution factor 1:100). Diluted cultures were shaken and incubated at 37°C in the platereader and $OD_{600}$ and YFP fluorescence was monitored every 10 min (YFP gain 120). YFP readings were normalized to $OD_{600}$ and averaged for each replicate at all timepoints at which $OD_{600}$ was between 0.20 and 0.25 (platereader units, that is, mid-exponential phase).

## Tetracycline resistance phenotyping on solid medium

Clones and strains to be tested were pregrown overnight in M9CG and diluted as shown in *Figure 2B* and *Figure 3—figure supplement 3*. We spotted 2.5 µL of diluted cultures on M9CG agar plates. After 24 hr incubation at 37°C, we took YFP fluorescence images of plates using a lab-

made macroscope (http://openwetware.org/wiki/Macroscope). The macroscope uses a Canon EOS 600D digital camera and a Canon EF-S 60 mm f/2.8 Macro USM lense (Canon, Tokyo, Japan). For illumination, we used a Cyan (505 nm) Rebel LED (Luxeon Star LEDs, Brantford, Canada) with a HQ500/20x excitation filter (Chroma, Bellow Falls, Vermont). As emission filter we used a camera-mounted D530/20 filter (Chroma).

## Reverse transcription

Stationary cultures of MG1655 Δ*tolC* (*rho*-wt) and of the isogenic strain with the *rho* M416I mutation in LB were diluted 1:100 in M9CG supplemented with tetracycline (0.44 µg/mL, that is, 50% of the MIC of strain MG1655 Δ*tolC* and grown overnight at 37°C with shaking. Total RNA was isolated using an Aurum Total RNA Mini kit (Bio-Rad) and DNA removed using an Ambion DNA-free kit (Life Technologies, Carlsbad, California). Isolated RNA was quantified using a Nanodrop spectrophotometer and integrity was checked on an agarose gel. cDNA was synthesized using an iScript cDNA synthesis kit (Bio-Rad) with 1 µg of total RNA as input in a 20 µL reaction. For the non-reverse-transcriptase (NRT) control reaction we used 0.5 µg of each of the two RNA samples.

## Endpoint PCR on cDNA

After reverse transcription, cDNA samples and the NRT control sample were diluted by adding 150 µL of nuclease-free water. Endpoint PCR to test for the presence of transcripts resulting from possible read-through across Rho-dependent terminators were done with a OneTaq Quick-Load Mastermix (New England Biolabs), using 1 µL of diluted cDNA or NRT control as template in a 50 µL reaction. To detect rare transcripts, we used 45 amplification cycles. As a positive control template in PCR reactions, we used 1 µL of a colony of strain MG1655 Δ*tolC* resuspended in 25 µL water and heated to 95°C for 4'. For agarose gel visualization, we loaded 15 µL of cDNA and NRT control PCR reactions and 2 µL of the positive control PCR reactions.

## Inferring the order of two adaptive mutations occurring in the same clone

In several cases, fluorescence analysis and sequencing revealed two potentially adaptive mutations in the same clone/population (colored dots on top of bars in *Figure 3A* and *Figure 5—figure supplement 1*). To infer which mutation came first, we proceeded as follows. For amplifications that occurred in combination with point mutations, we examined sequence chromatograms obtained from single clones. In all three cases, point mutations appeared as mixed nucleotide peaks, indicating that amplifications were initiated before the point mutations occurred. In two cases of amplifications co-occuring with upstream IS insertions, insertions occurred first. This is evident since PCR products used for sequencing appear as single bands of larger size than expected on agarose gels, whereas later insertions are expected to give two bands – a smaller one for copies without the insertion and a larger one for copies having the insertion. In one case, the insertion of IS3 upstream of locus *C* was a prerequisite for amplification initiation, as we could show by PCR that the IS3 insertion was at the amplicon junction. Cases of co-occurrence of amplifications with deletions or Rho-mutations were decided based on fluorescence trajectories. YFP/CFP ratios that remained high and relatively constant throughout the experiment indicate that amplifications expanded only after the other mutation had occurred. YFP/CFP ratios that increase at an intermediate timepoint during the experiment indicate that amplifications were initiated first. Last, we assume that a Rho mutation in strain *A* was selected only after the insertion of an upstream IS5 element, since *Figure 4B* indicates that Rho mutations alone would not have been adaptive in strain *A*. Rather, we assume that the Rho mutation enhanced transcriptional read-through from IS5 into the reporter cassette. Dots-on-bar color assignments in *Figure 3—figure supplement 5* do not reflect the order of mutations, as we did not do such analysis for additional replicate experiments.

## Assessment of gene essentiality

Essentiality data for upstream protein coding genes (*Figure 4A*) was taken from a published dataset (*Baba et al., 2006*). We did not find data on the essentiality of the *valU* tRNA operon upstream of locus *C* in the literature data. Therefore we tested experimentally, if deletions of the complete *valU* operon are tolerated. We attempted to delete the operon using recombineering with pKD13 as

template plasmid for a *kanR* cassette and primers valU_ko_f and valU_ko_r. The number of colonies on the *valU* knockout selection plate was more than tenfold lower than that of a control knockout of the neighboring *xapR* gene with primers xapR_ko_f and xapR_ko_r. To exclude that the low number of recombinants was due to a hairpin structure contained in the valU_ko_r primer, we repeated recombineering with a different reverse primer, valU_ko_r2, and obtained similar results. The low recombineering efficiency was not due to a smaller amount of PCR product used in transformations. Of six tested colonies obtained on the *ΔvalU::kanR* selection plate, only one colony gave a PCR product of the expected size in a test with flanking primers, showing that 5/6 colonies are not true *valU* knockouts. This suggests that *valU* deletion mutants require rare compensatory mutations to restore growth. Therefore the *valU* operon was considered as essential.

## Single-step plating experiments

We inoculated 1 mL of LB with a single colony of strains to be tested. After overnight incubation, saturated cultures were diluted 1:1000 into experimental evolution medium without tetracycline, and then split into 10 wells of a 96-well plate (220 μL / well). The 96-well plate was incubated on a plate shaker at 37°C for 24 hr to obtain saturated cultures, of which 180 μL containing approximately $2 \times 10^8$ cells were plated on M9CG medium with tetracycline at a concentration two times the MIC of respective strains (cell numbers were determined by plating dilutions on non-selective medium). Plates were incubated at 37°C in the dark and colonies counted every 24 hr. After 2 days, we picked one colony from every plate that had at least one colony on it and inoculated 200 μL of M9CG medium in a 96-well plate with the picked colony. After 24 hr incubation at 37°C, we used the VP407 pinner to spot approximately 2 μL on M9CG agar plates. After another 24 hr incubation, we took CFP fluorescence images of plates with the macroscope (see 'Tetracycline resistance phenotyping on solid medium'). For illumination, we used a Royal Blue (447.5 nm) Rebel LED (Luxeon Star LEDs) with a D436/20x excitation filter (Chroma). As emission filter we used a camera-mounted D480/40m filter (Chroma). The mean intensity of pixels of each spot was quantified. Spots with intensity six times greater than the mean intensity of all ancestor spots are considered to have amplifications (*Figure 6—figure supplement 2*).

## Statistical analysis

To test for homogeneity in the distribution of rescued vs. extinct populations, fluorescence phenotypes and mutation types, $r \times c$ Fisher's exact test for Count Data was used (fisher.test function in R [*R Core Team, 2012*]). For testing the distribution of mutation types, we used types indicated in *Figure 3A* by bar color, not dot color. For testing $2 \times 2$ contingency tables, Fisher's exact test was used with an alternative hypothesis of odds ratio $\neq 1$. Permutation tests were performed with the perm package (*Fay and Shaw, 2010*) for R (permTS function, method='exact.mc', $10^4$ Monte Carlo replications, two-sided).

## In-silico analysis of adaptive potential of *E. coli* gene neighborhoods (Venn diagram)

We used the Profiling of *E. coli* Chromosome (PEC) database available at https://shigen.nig.ac.jp/ecoli/pec/genes.jsp (accession number UA00096.2) and included all 4317 genes (feature type 'gene') of *E. coli* MG1655 with essentiality information in our analysis, which excludes non-coding genes. The position and orientation of promoters was extracted from Table S2 of the same study used to identify candidate transcripts in *Figure 4A* (*Conway et al., 2014*). We only included promoters annotated as 'primary' promoters in our analysis. The 'Promoter Confidence Score' was not taken into account. The position, orientation, and termination mode (intrinsic or non-intrinsic) of all terminators was extracted from Table S3 of the same study (*Conway et al., 2014*). In order to identify all genes downstream of Rho-dependent terminators (green circle in *Figure 7B*), we identified the closest upstream co-oriented terminator of every gene and evaluated whether it was predicted to be an intrinsic terminator or not, in which case we assumed it is Rho-dependent. In order to identify all genes to which a co-oriented upstream promoter could be co-opted by deletion without disrupting an essential gene, we first identified the next essential upstream gene of every gene, and then evaluated if there is at least one co-oriented promoter and intervening co-oriented terminator between the gene of interest and the next upstream essential gene. If this was the case, the gene of interest

was included in the respective set of genes (magenta circle in *Figure 7B*). In order to identify genes between flanking duplicates (blue circle in *Figure 7B*), we used the online REPuter tool (*Kurtz et al., 2001*) to find all forward repeats on the chromosome that satisfied the following criteria: repeat length $\geq$200 bp, Hamming distance $\leq$8, maximum distance between repeats 100 kb, minimum distance between repeats 200 bp. In this way, we identified four large regions of the MG1655 chromosome between flanking repeats: between IS1B and IS1C (containing 13 genes and locus *E*), between IS5H and IS5I (containing 42 genes and locus *B*), and between the ribosomal operons *rrnA* and *rrnC* (81 genes), and *rrnB* and *rrnE* (31 genes). We also obtained six genes between closely spaced repeats matching our criteria (*ybfB*, *ybfL*, *yibA*, *ldrA*, *ldrB*, and *ldrC*), which we did not include in the set 'between flanking duplicates', since the behavior of such closely spaced repeats might be different than those studied in our system. The Venn diagram was drawn in Matlab using the 'ChowRodgers' method for sizes of circles and intersection areas. A list of all included genes and their assignment to the three sets is available in *Figure 7—source data 1*.

### Data deposited online
Fastq files of whole genome sequencing of IS-free strain *D* (ancestor) and of evolved clones from four evolved populations of this strain (A11, C08, C10, D08) have been deposited in the IST Data Repository, 10.15479/AT:ISTA:65.

## Acknowledgements
We thank M Ackermann, N Barton, J Bollback, J Jäger, members of the Barton, Bollback, Bollenbach and Guet groups, and especially M Pleška for comments on earlier versions of the manuscript. We thank F Korč for assistance with the data analysis script.

## Additional information

### Funding
No external funding was received for this study.

### Author contributions
MS, Conceptualization, Data curation, Formal analysis, Investigation, Visualization, Methodology, Writing—original draft, Writing—review and editing; CCG, Conceptualization, Supervision, Funding acquisition, Methodology, Project administration, Writing—review and editing

### Author ORCIDs
Magdalena Steinrueck, http://orcid.org/0000-0003-1229-9719
Călin C Guet, http://orcid.org/0000-0001-6220-2052

## Additional files

### Supplementary files
• Supplementary file 1. Population trajectories. Set of 96-panel figures showing OD and OD-normalized fluorescence values for each population in each of 18 evolution experiments.

• Supplementary file 2. Source data populations. Excel table containing information on survival, fluorescence phenotypes, sequences, and mutation types of every experimental population, as well as information on which populations where used for further investigation (plasmid reconstruction etc.). This contains the source data of *Figures 2AC*, *3AB*, *5CD*, *Table 1*, and respective Figure Supplements.

• Supplementary file 3. Strains, plasmids, oligonucleotides. Excel table with all strains, plasmids and oligonucleotides used in this study

• Source code 1. Compressed file containing Matlab scripts and OD/YFP/CFP plate-reader raw data files of evolution experiments.

## Major datasets

The following dataset was generated:

| Author(s) | Year | Dataset title | Dataset URL | Database, license, and accessibility information |
|---|---|---|---|---|
| Steinrueck M, Guet CC | 2017 | Fastq files for "Complex chromosomal neighborhood effects determine the adaptive potential of a gene under selection" | http://dx.doi.org/10.15479/AT:ISTA:65 | Publicly available at the IST Data Repository (https://datarep.app.ist.ac.at/) |

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
