## [Decision Letter]

Thank you for submitting your article "Complex chromosomal neighborhood effects determine the adaptive potential of a gene under selection" for consideration by *eLife*. Your article has been reviewed by three peer reviewers, one of whom is a member of our Board of Reviewing Editors, and the evaluation has been overseen by Diethard Tautz as the Senior Editor. The reviewers have opted to remain anonymous.

The reviewers have discussed the reviews with one another and the Reviewing Editor has drafted this decision to help you prepare a revised submission.

Summary:

This paper uses a specially designed reporter to examine the nature of adaptation and the mutations that can drive tetracycline resistance under continual selection. The reviewers were generally enthusiastic about the work as such systematic analyses of mutational spectra during selection have the potential to inform us about evolution at a very detailed molecular level. In particular, this paper presents reasonably strong data to suggest that the local chromosomal context of a tet-resistance gene can influence its adaptation via differences in amplification potential (driven by IS elements) and read-through from other nearby genes. Despite some enthusiasm, the reviewers noted a few important weaknesses that would have to be remedied in a revised manuscript. These issues are detailed below and generally fall into three categories.

Essential revisions:

1) Some of the suggestions about how adaptation has occurred are incompletely substantiated. Additional experiments are needed to address these concerns; in most, if not all, cases, the necessary experiments are relatively straightforward.

– Results section; "All rescued populations displayed increased YFP fluorescence. Thus, rescue depended on the presence and overexpression of *tetA-yfp*." The second sentence doesn't logically follow from the first. The authors should test, at least in 1-2 cases, that selective deletion of TetA-YFP in the evolved clone eliminates rescue.

– "…or no upstream Rho-terminated transcripts were present (constraining adaptive Rho mutations, Figure 4)." The notion that the Rho mutations allow read-through into *tetA-YFP* is interesting, but the claim that no upstream Rho-terminated transcripts exist in strains A and C needs to be substantiated. Is this information being pulled from some prior study of Rho and if so, how reliable are these data? Or are they simply predictions based on Rho site bioinformatics? If it's the latter, then qRT-PCR data supporting the claims should be generated and shown.

– Figure 3: It wasn't clear how the evolved clones examined in a plasmid-based assay in Figure 3 were chosen. Additionally, it was unclear why 7 of the 14 mutations "reconstructed" on plasmids didn't recapitulate the evolved behavior. This point raises the important concern that the paper is ultimately about how genomic context can influence evolution, but the "reconstruction" experiments involve placing a limited genomic region onto a plasmid to drive YFP, not TetA-YFP, production. Could it be that 7 of the cases in Figure 3 failed because the genomic context has changed? This should be better discussed. In addition, the authors should, for at least a handful of the 'failed' cases in Figure 3, do proper reconstructions by engineering the putatively causal mutation into the ancestral strain background.

2) The presentation of results could be substantially improved. There is a general lack of primary data provided and many figures lack highly relevant details on the exact nature of the genomic context of each locus, both in the original strains and the evolved isolates.

– The authors should say something in the main text, not just the Materials and methods section, about (i) the rationale for choosing the four sites of insertion that they chose and (ii) the nature/genomic context of the four insertion sites. Genomic diagrams of the four insertion sites with flanking genes, etc should also be provided in the main Figure 1. One half of these regions is shown in Figure 4, but the other half is omitted and these diagrams should come earlier in the paper.

– "We verified the presence of IS5 at the boundary of the amplicon in rescued B populations (Figure 2)." Figure 2 appears to contain only a schematic of locus B, with no data supporting the claim that IS5 is at the boundary of the amplified region, nor is there really any primary data demonstrating or validating the notion that the cassette was amplified.

Subsection “Properties of upstream genes determine the availability of two different types of adaptive promoter co-option mutations”: The use of a strain that is apparently free of all IS elements needs more explanation. The strain generated in Posfai et al., 2006, has actually been reduced by 15% and so lacks a wide range of genomic features. This could complicate the interpretation that the difference in mutational spectra in the two strains (Figure 3) results from a lack of IS elements. The authors should more clearly discuss this issue and its impact on the interpretation of their results.

– In the same section: Without more details, it's difficult to assess or interpret the data in this section of the manuscript. It's not clear whether individual strains only have single mutations or multiple mutations and if it's the latter, were the individual mutations made? And while the nature of 'point mutations' is clear, it's not clear what the authors mean by mutations 'delivering full or partial promoters'. And in the cases where the reconstructed mutation does not increase the reporter, what's the explanation? Is it because there are other mutations?

3) There was a concern about whether the results depend on the precise way in which the selection was performed. The evolution is done with relatively small populations in microtiter plates that are diluted each day into fresh media whose tetracycline concentration increases by about 25% each day. Consequently, the populations are effectively in a race for increasing the expression of the tetA sufficiently fast to 'keep ahead' of the increasing tetracycline concentration. This suggests that even small differences in the initial expression levels of tetA at the different loci could have a large effect on the probability of survival. Thus, it seems plausible that the increased survival of clones with tetA at the B and D loci might be a result of higher initial expression at these loci. This might also explain puzzling observations like the fact that, in the IS-free case, rescue through point mutations in the promoter are observed 4 times at D but never at A or C, and the fact that rescue rates at the B locus vary across replicates even though CFP increase is observed almost without fail (Table 1). The authors should quantify initial expression levels at each locus and investigate the effect of this on the probability of rescue.

Related to this, the authors should include a panel of 96 figures (one for each well of the plate) showing the time courses of OD, YFP/OD, and CFP/OD across the 10 days. Right now the authors only report numbers of rescued and numbers with YFP or YFP+CFP increase, but it is hard to judge if such a simple classification does justice to the potentially complex dynamics that is observed across populations.

Apart from the effect of initial expression, to what extent does the selection for 'fast increase in expression' bias the mutations that are observed to one type or another? To argue for the generality of the observations that the authors make, it would be useful to assess the rate of rescue in another setting where the type of selection is different, e.g. using a Luria-Delbrueck type experiment in which many cultures (with tetA at the different loci) are grown in media without tetracycline, with colonies counted on agar plates with tetracycline concentrations beyond the MIC. Does one again find mostly rescue of clones at B and D, even in the IS free case?

[Editors' note: further revisions were requested prior to acceptance, as described below.]

Thank you for submitting your article "Complex chromosomal neighborhood effects determine the adaptive potential of a gene under selection" for consideration by *eLife*. Your article has been reviewed by two peer reviewers, one of whom is a member of our Board of Reviewing Editors, and the evaluation has been overseen by Diethard Tautz as the Senior Editor. The reviewers have opted to remain anonymous.

The reviewers have discussed the reviews with one another and the Reviewing Editor has drafted this decision to help you prepare a revised submission.

Summary:

The reviewers are still generally positive and there was a consensus that no additional experiments or analyses are critical at this stage. However, there was a continued concern, also raised in the first round of review, that the Introduction is somewhat disorienting and excessively abstract. While the reviewers appreciate the effort to place the work in a broad and general context, they also felt strongly that the paper would benefit from a more direct, concrete exposition of what specific questions were addressed, what was done, and what was learned. The use of simple, straightforward language is encouraged. It's not a matter of "dumbing" down the Introduction, but making it clear and accessible. The comments of one reviewer are provided in full below as they will be helpful as you further revise the manuscript.

Essential revisions:

As I suspected the initial MIC's of the different constructs (cassette at locus A, B, C or D) are not the same, but even the initial B and D populations already have elevated MIC. The authors explain that they attempt to normalize for this difference by treating each construct with a concentration profile that is the same relative to the original MIC of that construct. That is, the constructs at locus D will be subjected to time-dependent anti-biotic concentrations that are about twice that of what the construct at locus A are subjected to (based on data presented in Figure 1—figure supplement 1).

I understand that this is the most reasonable way to correct for the differences in MIC of the initial populations, but I am not really convinced that it will now make the selective environments directly comparable. That is, it seems likely that, even if no beneficial mutations were to occur at all, substantial differences in the population sizes would remain and these might substantially affect the probability of generating beneficial mutants (i.e. just because of an elevated population size).

I think it would be good for the authors to add some discussion of this issue. I personally feel this issue is the main motivation for doing the Luria-Delbrueck like experiments (i.e. to check that the results are not specific to the precise way of performing the selection) and it might be worthwhile to mention this also.

Finally, regarding this point, although I feel that it still cannot be definitively said that the increased rescue of B and D constructs is solely due to the increased RATE of generating beneficial mutations rather than an increase in the size of the target population for such mutations, what one could definitely conclude is that, if this cassette were to be inserted in the genome through horizontal transfer, than rescue under selection with antibiotic is clearly highest when it is inserted into loci B or D, rather than at A or C.

The other comment I have is about presentation. I still find the introductory sections rather disorienting regarding what this paper is going to actually show. My impression is that this is due to a choice of presentation style which seems to aim to present the questions that the work addresses in as general and abstract terms as possible, leaving it typically up to the reader to connect the general abstract description to the actual concrete things that are being done. I find this rather tiresome to read. I believe the paper would become much more pleasant to read if the authors simply removed all attempts at phrasing large general abstract questions about evolution, and instead just say what exactly they do and what they find. The readers will then themselves immediately appreciate the generality of these findings.

(As an aside, I found the start of paragraph two in subsection “Different mutation types and their rate biases” a particularly striking example of general abstract phrasing that is almost impossible to parse, e.g. I have no idea what it means for mutation types to sustain adaptations.)

In the end it seems to me that the authors have a rather straight-forward and simple message: The rate at which genetic mutations occur that increase the expression level of a gene depends strongly on features of the genomic context of the gene, i.e. the rate at which the region is duplicated, the rate at which deletions or insertions cause promoter sequences to be placed upstream of the gene, and the rate of mutations causing read-through from neighboring genes. In contrast, apart from the read-through mutations, the EFFECT of the observed mutations, once introduced, is largely independent of this chromosomal context (as the experiments with the plasmids show).

Why not stress rather than obscure that these are the concrete results?

---

## [Author Response]

Essential revisions:

*1) Some of the suggestions about how adaptation has occurred are incompletely substantiated. Additional experiments are needed to address these concerns; in most, if not all, cases, the necessary experiments are relatively straightforward.*

– Results section; "All rescued populations displayed increased YFP fluorescence. Thus, rescue depended on the presence and overexpression of tetA-yfp." The second sentence doesn't logically follow from the first. The authors should test, at least in 1-2 cases, that selective deletion of TetA-YFP in the evolved clone eliminates rescue.

We agree that the logic of this sentence in isolation is incomplete. We now modified this sentence and combined it with the previous one to clarify that in the absence of the reporter cassette, rescue was never observed. We also performed the requested experiments: Figure 2 shows how resistance to tetracycline is lost in 3/3 cases in which we selectively deleted *tetA-yfp* from evolved clones derived from three different ancestral strains.

– "…or no upstream Rho-terminated transcripts were present (constraining adaptive Rho mutations, Figure 4)." The notion that the Rho mutations allow read-through into tetA-YFP is interesting, but the claim that no upstream Rho-terminated transcripts exist in strains A and C needs to be substantiated. Is this information being pulled from some prior study of Rho and if so, how reliable are these data? Or are they simply predictions based on Rho site bioinformatics? If it's the latter, then qRT-PCR data supporting the claims should be generated and shown.

To our knowledge, there is no reliable way of inferring Rho termination sites using bioinformatics, because they show almost no conservation at the sequence level ((Ciampi, 2006), cited also in the revised main text) and only a handful of Rho-termination sites have been studied in detail. Rather, candidate Rho termination sites on a genomic level are found by the absence of signatures of intrinsic termination sites (stem-loop structures), which are more readily identifiable. The RNAseq study from which we drew the information on termination sites as shown in Figure 4 (Conway et al., 2014) used this approach, and combined experimental evidence for the end of a transcript (single-bp RNAseq) with the commonly used bioinformatic algorithm TransTermHP (Kingsford, Ayanbule, and Salzberg, 2007) to detect intrinsic termination sites. Importantly, the level and termination of transcripts can be condition-dependent, and our experimental conditions are not exactly the same as in the RNAseq study. As explained in the revised text and shown in Figure 4, we therefore also generated direct RNA-level data to verify that there are transcripts spanning putative Rho-termination sites upstream of loci *B* and *D* in a way that is dependent on the Rho status (i.e. elevated transcript levels across the candidate terminator transcripts in the Rho mutant compared to the Rho wt). While we cannot completely rule out the presence of Rho-terminated active transcripts upstream of loci *A* and *C*, we did not find evidence for them at the level of RNA (Figure 4) or at the phenotypic level (Figure 4). As suggested by the reviewers, we also attempted to obtain RT-qPCR data to quantify RNA levels directly at candidate terminator sites upstream of all four loci, however we found this data difficult to interpret because of three reasons: (1) Despite optimization attempts, several of the qPCR primer pairs had very poor amplification efficiencies, likely because they spanned stem-loop structures that are typical of intrinsic termination sites. (2) Given the potentially global effect of Rho mutations on the transcriptome (Freddolino, Goodarzi, and Tavazoie, 2012), it is not obvious which genes could be used as valid reference genes to compare transcript levels in Rho-wt vs. mutant backgrounds. (3) Standard RT-qPCR data using small amplicons does not allow to draw conclusions about the directionality of read-through at two head-to-head facing terminators (see Figure 4—figure supplement 1 for such configurations), and it does not give information on whether read-through propagates all the way downstream to the reporter cassette insertion loci. Importantly, even in the absence of quantitative data on Rho-dependent read-through at terminators upstream of loci *A-D*, our interpretation is now supported by four lines of evidence: (1) Position and termination mode of transcripts and terminators as found in the RNA-seq study (Figure 4). (2) The integration-locus-dependent effect of Rho mutations on tetracycline resistance, which is the phenotype under selection in our evolution experiments (Figure 4). (3) RNA-level evidence (Figure 4). (5) Restriction of Rho mutations to strains with the reporter cassette at loci *B* and *D* (Figure 3, disregarding one Rho mutation in strain *A* which co-occurred with an upstream IS5 insertion that changes the upstream neighborhood at locus *A*).

– Figure 3: It wasn't clear how the evolved clones examined in a plasmid-based assay in Figure 3 were chosen.

We agree that the section on mutation reconstructions on plasmids was originally written in a somewhat confusing way. We now placed this information in a separate section ‘Adaptation involves a broad variety of different mutation types’, which we expanded in order to clarify how and why we chose the mutations that we reconstructed, and how to interpret the results from reporter plasmids. Since in this part, we were interested in context-independent effects of mutations (hence the reconstruction on plasmids), we only reconstructed mutations that affected the sequence of *p_0_*, which is common to all strains. This included point mutations and a small deletion with *p_0_* (Figure 3), as well as IS insertions into *p_0_* (Figure 3). We only reconstructed mutations found in the first set of evolution experiments (i.e. those shown in Figure 3). In [Supplementary-material SD15-data], which contains information on every replicate population of every evolution experiment, we included a column ‘Reconstructed on reporter plasmid’ for easy identification of these mutations.

Additionally, it was unclear why 7 of the 14 mutations "reconstructed" on plasmids didn't recapitulate the evolved behavior. This point raises the important concern that the paper is ultimately about how genomic context can influence evolution, but the "reconstruction" experiments involve placing a limited genomic region onto a plasmid to drive YFP, not TetA-YFP, production. Could it be that 7 of the cases in Figure 3 failed because the genomic context has changed? This should be better discussed.

The issue of context-dependent mutation effects is indeed very central, and we tried to clarify it better throughout the text. We think that the particular concern of the reviewers here stems from a confusion rooted in the rather misleading way in which Figure 3 and its discussion in the text was originally presented. We split up Figure 3 of the original submission into three panels (Figure 3) to aid the understanding of this figure. Figure 3 shows plasmid reconstructions of 6 unique mutations within *p_0_* (one of them a small deletion). Figure 3 shows reconstructions of four different IS sequences that we had found inserted into *p_0_* (IS5 at two different insertion points). For Figure 3, we cloned 600 bp of the four IS directly upstream of the *yfp*-reporter to quantify effects on expression independent of the *p_0_* insertion context. Figure 3 therefore show 10 primary mutations (not 14) analyzed in these experiments. 8 out of 10 of these reconstructed mutations did increase reporter fluorescence significantly. In the revised text, we discuss why the lack of reporter fluorescence from one point mutation (T-145C) does not represent a ‘failed’ reconstruction. The ‘failure’ of 1-2 IS insertion reconstruction is addressed in Figure 3, which is a follow-up of Figure 3. In Figure 3, we investigate those IS insertions that did not increase *yfp* expression on plasmids when cloned directly upstream of the reporter (IS1 and IS5). For Figure 3, we engineered the terminal 50 bp of these two IS within the *p_0_* sequence at the precise point where we had found them inserted in evolved clones (hence, two reconstructions for IS5). For IS1, we observe reporter fluorescence that depends on the precise insertion point, which indicates a hybrid promoter at the insertion point. For IS5, we did not find evidence of a hybrid promoter. The ‘failure’ of the IS5 reconstruction is discussed in the main text and addressed in the newly included Figure 3—figure supplement 3 (see our reply to the next point).

In addition, the authors should, for at least a handful of the 'failed' cases in Figure 3, do proper reconstructions by engineering the putatively causal mutation into the ancestral strain background.

As explained above, only one of the 10 tested mutations, the insertion of IS5 upstream of the reporter cassette can be considered as ‘failed’ in the sense of lacking reporter fluorescence on the plasmid reconstruction. We engineered one of the IS5 insertions from an evolved clone of strain *A* into the ancestral background and showed that it restores growth on tetracycline and *tetA-yfp* fluorescence. Thus, the activating effect of IS5 on downstream gene expression appears to depend indeed on the chromosomal context. The respective data is shown in Figure 3—figure supplement 3 and explained in the Results section.

2) The presentation of results could be substantially improved. There is a general lack of primary data provided and many figures lack highly relevant details on the exact nature of the genomic context of each locus, both in the original strains and the evolved isolates.

– The authors should say something in the main text, not just the Materials and methods section, about (i) the rationale for choosing the four sites of insertion that they chose and (ii) the nature/genomic context of the four insertion sites. Genomic diagrams of the four insertion sites with flanking genes, etc should also be provided in the main Figure 1. One half of these regions is shown in Figure 4, but the other half is omitted and these diagrams should come earlier in the paper.

We moved the section on the choice of the loci to the beginning of the Results, in which we explain our experimental system. We also corrected an error on the rational for choosing locus *C*. Genomic diagrams showing flanking genes both up- and downstream of the insertion loci are now shown in main Figure 1. The chromosomal neighborhood at a larger scale with the position of insertion sequences that could provide flanking homology is shown in Figure 3—figure supplement 1, which also shows the regions in which the IS-free strains based on strain MDS42 differs from the MG1655-based ‘IS-wt’ strains.

– "We verified the presence of IS5 at the boundary of the amplicon in rescued B populations (Figure 2)." Figure 2 appears to contain only a schematic of locus B, with no data supporting the claim that IS5 is at the boundary of the amplified region, nor is there really any primary data demonstrating or validating the notion that the cassette was amplified.

We included an agarose gel as part of Figure 2, which shows the presence of the PCR product corresponding to the expected IS5I/H junction in evolved populations with increased CFP fluorescence, and the absence of the junction in the ancestral genotype and amplified populations from strain *B∆IS5I*, in which one of the flanking IS is missing.

We validated the amplification of the reporter cassette in populations with increased CFP fluorescence by qPCR on genomic DNA extracted from several populations evolved from different ancestor strains. We agree that using CFP as a marker of amplification is a central pillar of this study and that we needed to make more clear how it was validated. Therefore, for the revised version, we moved the respective figure that shows a correlation between CFP fluorescence and genomic DNA copy number from the supplement to main Figure 1. The respectively extended part of the revised text is now at the end of the first section in the Results.

Subsection “Properties of upstream genes determine the availability of two different types of adaptive promoter co-option mutations”: The use of a strain that is apparently free of all IS elements needs more explanation. The strain generated in Posfai et al., 2006, has actually been reduced by 15% and so lacks a wide range of genomic features. This could complicate the interpretation that the difference in mutational spectra in the two strains (Figure 3) results from a lack of IS elements. The authors should more clearly discuss this issue and its impact on the interpretation of their results.

Posfai et al. performed several different tests on mutation rates and spectra of strain MDS41, a close relative of MDS42, the parent strain of our IS-free strains, and compared mutation rates to MG1655, the parent strain of our IS-wt strains. They concluded that the difference in the total mutation rate between the two strains is fully explained by the lack of IS-mediated mutations in MDS41; mutations of all other tested types were found to occur at similar rates in both strains. We now explain this at the beginning of the new Results section “Adaptation involves a broad diversity of mutation types”. We also added Figure 3—figure supplement 1 to show which regions in the larger neighborhood of our insertion loci have been deleted in the MDS42 background. We cannot exclude that the deletions in the MDS42 background have effects on adaptation in our experiments that we are not aware of. For example, deletions may affect regulation of genes in the neighborhood of our loci. Therefore, we have avoided quantitative comparisons of adaptation between IS-wt and IS-free strains. Rather, given that the only difference within the sets of IS-wt and IS-free strains is the integration locus of the reporter cassette, we interpret differences in adaptation within these sets of strains as indicators of neighborhood effects. We then compare the differences found in the set of IS-wt strains and in the set of IS-free strains to rationalize whether these differences are dependent on the local neighborhood (as is the case for promoter co-opting mutations which are locus-specific in both sets) or on the more global neighborhood (as is the case for amplifications mediated by flanking homologous IS elements).

– In the same section: Without more details, it's difficult to assess or interpret the data in this section of the manuscript. It's not clear whether individual strains only have single mutations or multiple mutations and if it's the latter, were the individual mutations made? And while the nature of 'point mutations' is clear, it's not clear what the authors mean by mutations 'delivering full or partial promoters'. And in the cases where the reconstructed mutation does not increase the reporter, what's the explanation? Is it because there are other mutations?

As explained in our reply on mutation reconstructions on plasmids (see above), we have largely expanded the section in the Results to clarify these questions raised be the reviewers. Multiple mutations found in the same clone are indicated by colored dots on bars shown in Figure 3, some of these multiple mutations are explained at respective points in the text (e.g. mutations that occur with amplifications are investigated in the section “Chromosomal neighborhood influences adaptation by affecting the fitness cost of amplifications” of the Results). By mutations ‘delivering full or partial promoters’ we mean IS elements that contain promoters, which can drive expression of genes downstream of their insertion point (IS2, IS3), or that contain the upstream part of promoters which require a corresponding sequence downstream at the point of insertion (IS1) (Figure 3). Cases ‘where the reconstructed mutation does not increase the reporter’ are discussed above in our reply to the last point of part (1) of the decision letter. After our initial sequencing (of clones from rescued populations from IS-wt strains of the first replicate experiment set), which only included the upstream sequence of the *tetA-yfp* gene, we did have four unexplained cases of adaptations without a known candidate causative mutation. We addressed these four cases with whole-genome sequencing, and discovered Rho mutations in three of them – as explained at the beginning of the section “Properties of upstream genes determine the availability of two different types of adaptive promoter co-option mutations” and in Figure 3—figure supplement 4. For all other experiments (IS-free strains and additional replicate experiment sets), candidate adaptive mutations were then found using Sanger sequencing of the *tetA-yfp* upstream region and the *rho* gene. In this way, we found a candidate adaptive mutation for every rescued population, with a few exceptions that we did not follow up on.

3) There was a concern about whether the results depend on the precise way in which the selection was performed. The evolution is done with relatively small populations in microtiter plates that are diluted each day into fresh media whose tetracycline concentration increases by about 25% each day. Consequently, the populations are effectively in a race for increasing the expression of the tetA sufficiently fast to 'keep ahead' of the increasing tetracycline concentration. This suggests that even small differences in the initial expression levels of tetA at the different loci could have a large effect on the probability of survival. Thus, it seems plausible that the increased survival of clones with tetA at the B and D loci might be a result of higher initial expression at these loci. This might also explain puzzling observations like the fact that, in the IS-free case, rescue through point mutations in the promoter are observed 4 times at D but never at A or C, and the fact that rescue rates at the B locus vary across replicates even though CFP increase is observed almost without fail (Table 1).

Our understanding of the evolutionary rescue scenario (‘adapt or die out’) faced by evolving populations matches exactly what the reviewers describe as a ‘race for increasing the expression of *tetA-yfp* sufficiently fast to ‘keep ahead’ of the increasing tetracycline concentration. We now included a sentence at the beginning of the Results section that explicitly states this: ‘Rescue from extinction requires the spread of adaptive mutations in a race against population decline.’ We also agree with the reviewers that the initial level of *tetA-yfp* expression is expected to have a large effect on survival, as it sets the decline rate of the unadapted genotype (see (Martin et al., 2013) for an accessible description of the mathematical framework used to describe evolutionary rescue and the impact of peculiarities of evolution experiments such as periodic bottlenecking). Since we did not want to study the effect of chromosomal location on initial expression (for which our experiments would have been a very complicated way to do so), but the effect of chromosomal location on adaptive mutations that can change expression, we designed our experiments to account for initial expression differences as best as we can. In the revised text, we have made it more clear why and how exactly we did so. The respective section at the beginning of the Results starts with the sentence “The probability of evolutionary rescue depends on the size and decline rate…”.

Concerning the numbers of rescue by point mutations observed in different IS-free strains, numbers are too low to draw conclusions supported by statistics.

We agree that the variability in the number of rescued populations of strain *B* between replicate experiments is likely an effect of fluctuations in the relatively small population size at the time of daily transfer. Although we did not quantify this explicitly, we think that the largest component causing such fluctuations comes from variability in the amount of liquid (around 2 µL) transferred with the pin inoculator that we used (and of course from the inherent stochasticity of mutations). We mention this also in the revised text (at the end of the section “The chromosomal location of a selected gene has large effects on adaptation” in the Results section).

The authors should quantify initial expression levels at each locus and investigate the effect of this on the probability of rescue.

Since initial expression levels of *tetA-yfp* are very low (although maybe surprisingly not zero, as might be expected from a randomized upstream sequence as *p_0_*), we could not use YFP fluorescence to quantify initial expression. What is more relevant in our experiments, and apparently also much more sensitive, are differences in tolerance to tetracycline between strains. We quantified these differences by measuring dose-response curves of each strain to tetracycline at a high resolution of tetracycline concentration (Figure 1—figure supplement 1, to which we added explicit MIC values for each strain in the revised version). We consider differences in the MIC as a good proxy for initial expression levels at each locus, given that apart from the integration locus strains are isogenic. To account for initial expression levels in evolution experiments, we adjusted tetracycline concentrations to the initial MIC levels. We made this also more explicit in the main text (starting with “Given the otherwise isogenic background of strains, …”).

Related to this, the authors should include a panel of 96 figures (one for each well of the plate) showing the time courses of OD, YFP/OD, and CFP/OD across the 10 days. Right now the authors only report numbers of rescued and numbers with YFP or YFP+CFP increase, but it is hard to judge if such a simple classification does justice to the potentially complex dynamics that is observed across populations.

The requested figures are now found in [Supplementary-material SD14-data], which we also refer to in the main text. Certainly, adaptation dynamics are more complex than suggested by the fluorescence phenotype classification and time-resolved fluorescence data across the 10 days can be informative about these dynamics. Since we have sequencing data however only for the endpoint of the experiment (day 10), interpreting these dynamics is hard. We therefore decided to analyze the temporal dynamics of fluorescence only in relation to the specific questions that came up during our investigation and that could be addressed with it (i.e. the section on amplification cost).

Apart from the effect of initial expression, to what extent does the selection for 'fast increase in expression' bias the mutations that are observed to one type or another? To argue for the generality of the observations that the authors make, it would be useful to assess the rate of rescue in another setting where the type of selection is different, e.g. using a Luria-Delbrueck type experiment in which many cultures (with tetA at the different loci) are grown in media without tetracycline, with colonies counted on agar plates with tetracycline concentrations beyond the MIC. Does one again find mostly rescue of clones at B and D, even in the IS free case?

Clearly, the precise way in which selection is imposed should matter for the outcome of the experiment. We mention this rather early in the Results section (“… illustrating how our experimental selection filters for mutations that increase *tetA-yfp* expression above a minimum level.”). It is known that the precise way in which conditions deteriorate can affect the genetic architecture of adaptations, by making particular routes to adaptation inaccessible (Lindsey et al., 2013). In our particular experiment, the gradual increase in tetracycline concentration is expected to favor amplifications, which according to the ‘canonical’ model are built up in a sequential process that starts from an initial duplication (Romero and Palacios 1997). We included a new section and Figure (‘Chromosome neighborhood effects on adaptation in a single-step plating experiment’ and Figure 6), in which we present the results of plating experiments as suggested by the reviewers. Importantly, since selection on tetracycline plates is not stringent, i.e. sensitive clones may grow at very low rates even at tetracycline concentrations above the MIC, the results of these experiments should not be interpreted in the same way as the classical Luria-Delbrueck fluctuation experiment. This is because mutations may occur also after plating, which may cause a misinterpretation of mutation rates calculated using colony counts (Foster 2006). However if colonies are counted at the earliest possible time point (Day 2 in Figure 6), we expect colony numbers to report mostly on the locus-specific rate of single-step mutations, i.e. point mutations, IS insertions, and promoter co-option mutations. At later time points (Day 5 in Figure 6), multi-step adaptations such as amplifications (and possibly other multi-step adaptations) become dominant. The dynamics of colony appearance in our plating experiment recapitulates several aspects of Cairns’ classic *lacZ* frameshift reversion experiment (Cairns, Overbaugh, and Miller 1988), which has been linked to transient amplification of the selected gene (Andersson 1998) in a process that heavily depends on the position of the *lacZ* gene in a way similar to our system (Slechta et al., 2002). Regardless of the temporal dynamics of colony appearance, rescued clones are found mostly with the reporter cassette at loci *B* and *D*, also in the IS-free case, which recapitulates the results from our evolution experiments in liquid culture.

[Editors' note: further revisions were requested prior to acceptance, as described below.]

Summary:

The reviewers are still generally positive and there was a consensus that no additional experiments or analyses are critical at this stage. However, there was a continued concern, also raised in the first round of review, that the Introduction is somewhat disorienting and excessively abstract. While the reviewers appreciate the effort to place the work in a broad and general context, they also felt strongly that the paper would benefit from a more direct, concrete exposition of what specific questions were addressed, what was done, and what was learned. The use of simple, straightforward language is encouraged. It's not a matter of "dumbing" down the Introduction, but making it clear and accessible. The comments of one reviewer are provided in full below as they will be helpful as you further revise the manuscript.

We thank the reviewers and in particular the author of the below review for their specific criticism and their concrete suggestions, which we implemented as detailed in the following.

We largely restructured the Introduction and removed most of the opening paragraph. To improve readability and focus, we also shortened the section on different mutation types and their position biases. The last paragraph of the Introduction now contains a specific exposition of our research question, the methodology, the results and the conclusions.

Essential revisions:

As I suspected the initial MIC's of the different constructs (cassette at locus A, B, C or D) are not the same, but even the initial B and D populations already have elevated MIC. The authors explain that they attempt to normalize for this difference by treating each construct with a concentration profile that is the same relative to the original MIC of that construct. That is, the constructs at locus D will be subjected to time-dependent anti-biotic concentrations that are about twice that of what the construct at locus A are subjected to (based on data presented in Figure 1—figure supplement 1).

Yes, this is correct.

I understand that this is the most reasonable way to correct for the differences in MIC of the initial populations, but I am not really convinced that it will now make the selective environments directly comparable.

Agreed. The correction for initial MIC differences is no direct correction for population size differences throughout the experiment. The correction rather addresses the otherwise large population size differences at the beginning of the experiment, making selective environments as comparable as possible in a practical way. We have added an explanatory sentence on the purpose of the MIC correction in the caption of Figure 1—figure supplement 1.

That is, it seems likely that, even if no beneficial mutations were to occur at all, substantial differences in the population sizes would remain and these might substantially affect the probability of generating beneficial mutants (i.e. just because of an elevated population size).

I think it would be good for the authors to add some discussion of this issue.

We now discuss this possibility in the section on the Luria-Delbrueck like experiments. We have also modified a sentence earlier in the Results section, which previously may have implied that results from strains are perfectly comparable.

I personally feel this issue is the main motivation for doing the Luria-Delbrueck like experiments (i.e. to check that the results are not specific to the precise way of performing the selection) and it might be worthwhile to mention this also.

We have now explicitly mentioned the above concern as a motivation for the Luria-Delbrueck like experiments and explain how these experiments address this potential problem.

Finally, regarding this point, although I feel that it still cannot be definitively said that the increased rescue of B and D constructs is solely due to the increased RATE of generating beneficial mutations rather than an increase in the size of the target population for such mutations, what one could definitely conclude is that, if this cassette were to be inserted in the genome through horizontal transfer, than rescue under selection with antibiotic is clearly highest when it is inserted into loci B or D, rather than at A or C.

Yes, this conclusion is definitely valid. We have now included it in the Discussion along with the relevance of our results for horizontal gene transfer.

The other comment I have is about presentation. I still find the introductory sections rather disorienting regarding what this paper is going to actually show. My impression is that this is due to a choice of presentation style which seems to aim to present the questions that the work addresses in as general and abstract terms as possible, leaving it typically up to the reader to connect the general abstract description to the actual concrete things that are being done. I find this rather tiresome to read. I believe the paper would become much more pleasant to read if the authors simply removed all attempts at phrasing large general abstract questions about evolution, and instead just say what exactly they do and what they find. The readers will then themselves immediately appreciate the generality of these findings.

As explained in our reply to the reviewing editor’s summary, we have removed the respective parts of the Introduction and added a specific summary of our research question and our results at the end of the Introduction.

(As an aside, I found the start of paragraph two in subsection “Different mutation types and their rate biases” a particularly striking example of general abstract phrasing that is almost impossible to parse, e.g. I have no idea what it means for mutation types to sustain adaptations.)

We removed this sentence.

In the end it seems to me that the authors have a rather straight-forward and simple message: The rate at which genetic mutations occur that increase the expression level of a gene depends strongly on features of the genomic context of the gene, i.e. the rate at which the region is duplicated, the rate at which deletions or insertions cause promoter sequences to be placed upstream of the gene, and the rate of mutations causing read-through from neighboring genes. In contrast, apart from the read-through mutations, the EFFECT of the observed mutations, once introduced, is largely independent of this chromosomal context (as the experiments with the plasmids show).

Why not stress rather than obscure that these are the concrete results?

Understood. Our main message is now summarized in the last paragraph of the Introduction of our manuscript. We thank the reviewer for these specific suggestions, which helped make our manuscript more accessible.